# Composition of Rare Earth Elements in Fluvial Sediments of the Lesser Zab River Basin, Northeastern Iraq: Implications for Tectonic Setting and Provenance

**Younus I. Al-Saady** [1], **Arsalan Ahmed Othman** [2,3,*], **Yousif O. Mohammad** [4], **Salahalddin S. Ali** [5], **Sarmad A. Ali** [6,7], **Veraldo Liesenberg** [8] and **Syed E. Hasan** [9]

1   Iraq Geological Survey, Al-Andalus Square, Baghdad 10068, Iraq; younusalsaady@gmail.com or younusalsaady@geosurviraq.iq
2   Iraq Geological Survey, Sulaymaniyah Office, Sulaymaniyah 46013, Iraq
3   Department of Petroleum Engineering, Komar University of Science and Technology, Sulaimaniyah 46013, Iraq
4   Department of Geology, College of Science, University of Sulaimani, Sulaimaniyah 46013, Iraq; yousif.mohammad@univsul.edu.iq
5   Civil Engineering Department, College of Engineering, Komar University of Science and Technology, Sulaimaniyah 46013, Iraq; salah.saeed@komar.edu.iq
6   Department of Applied Geology, College of Science, Kirkuk University, Kirkuk 36013, Iraq; sarmad@uokirkuk.edu.iq or sarmad@uow.edu.au
7   GeoQuEST Research Centre, School of Earth, Atmospheric and Life Sciences, University of Wollongong, Wollongong, NSW 2522, Australia
8   Department of Forest Engineering, Santa Catarina State University, Lages 88520-000, Brazil; veraldo.liesenberg@udesc.br
9   Department of Earth & Environmental Sciences, School of Science and Engineering, University of Missouri, Kansas City, MO 64110-2499, USA; hasans@umkc.edu
*   Correspondence: arsalan.aljaf@gmail.com or arsalan.aljaf@geosurviraq.iq or arsalan.aljaf@komar.edu.iq

**Abstract:** During the past few decades, rare earth elements (REEs) have gained enormous attention in geochemical studies worldwide as a result of their important role in the manufacturing of high-tech equipment. REEs in river sediment have been widely used for provenance determination and in geochemical studies of continental crust, rock and sediment environments, and anthropogenic pollution. This study aims to elucidate the origin and tectonic setting of Little Zab River Basin (LZRB) sediments by examining 23 fluvial sediment samples of rare earth elements (REEs) collected from both the primary river and the inter-sub-basin regions during the rainy or high-flow season. The ICP-MS method was employed to analyze all samples to identify and assess the compositions of REEs. A fraction of the river sediments, smaller than 2 mm, which is more representative and more homogeneous, was used to carry out geochemical analysis. REE concentrations in the Little Zab River (LZR) and the upper parts of the LZRB were generally higher than those in the lower parts. The concentration of REEs in nearly all samples was lower than that of the North American Shale Composite (NASC), and the Upper Continental Crust (UCC), except for the sub-basin sediment Sbs2, which was higher than these references; also, the sediment sample Zrs4 was slightly higher than NASC. Light rare earth elements (LREEs) display enrichment relative to heavy rare earth elements (HREEs) with a range between 7.15 μg/g and 12.37 μg/g for LZR samples and between 5.95 μg/g and 13.03 μg/g for the sub-basin samples. The REE discrimination diagrams, along with the chondrodite-normalized pattern of the studied sediments, confirm that the sediment is predominantly sourced from the alkaline basaltic unit of the late Cretaceous Walsh group of an arc tectonic affinity.

**Keywords:** rare earth elements; REE; Little Zab River Basin; fluvial sediments; tectonic setting; Iraq

## 1. Introduction

Rare earth elements (REEs) are a group of elements from lanthanum (La) to lutetium (Lu), which are enriched or depleted based on mineral contents and physical and chemical processes, and have received much attention in the past decades [1]. Their chemical characteristics gradually change with increasing ionic radii, resulting in a slight variation in behavior during weathering, transportation, and precipitation; they also have high resistance to chemical mobilization [2,3]. REE concentrations are minimal in minerals that form during the initial phase of magma crystallization, which is dominated by olivine and pyroxene. However, they become more pronounced as accessory minerals that crystallize later in the magma's cooling process [4]. Additionally, these accessory minerals, such as monazite, allanite, sphene, apatite, and zircon separate the REE into different fractions. For instance, garnet preferentially incorporates the heavy rare earth elements (HREEs) over the light rare earth elements (LREEs) [5,6]. Due to their high charges and large ionic sizes, REEs cannot replace the primary constituents of common minerals found in igneous rock [6]. The abundance of REEs in rivers depends mainly on the lithology of the drainage basin and bedrock [7]. Throughout the weathering process of rocks, REEs can be sequestered or immobilized through various mechanisms. These mechanisms include: (i) retention within primary minerals resistant to weathering, (ii) integration into newly formed crystalline or amorphous mineral phases, and (iii) adsorption by clays. The mobilization and redistribution of lanthanides may be heightened by the accelerated dissolution of specific primary REE minerals, such as zircon, xenotime, apatite, and feldspar, as a result of reactions with weathering agents [8]. As rare earth elements (REE) often exhibit low solubility in surface environments, levels of REEs found in sedimentary deposits primarily mirror the geological composition of the source area. Consequently, patterns of REE in river sediments serve not only to pinpoint economically viable deposits but are also commonly employed in provenance studies that do not have a direct economic emphasis [9,10].

The Lesser or Little Zab River (LZR), one of the major tributaries of the Tigris River, is the largest drainage basin within the Iraqi borders and respresents the main source of water supply. Several factors influence the mobilization, fractionation, and composition of REEs in sediments [8,11]. Due to the large (~20,000 km$^2$) area of the LZR drainage basin and the presence of a wide variety of rock types, it is difficult to precisely determine the characteristics of source rocks [12,13]. Mixing of REEs during denudation, transportation, and deposition results in homogenous REE patterns in large rivers [13]. Ferhaoui et al. reveal that there is no significant difference in REE concentrations between different grain-size fractions of sediments [4]. However, [14] suggested removing coarse debris and gravel from the sediments to be more representative and more homogeneous of the bulk. Several published studies indicate that the fractionation of REEs occurs during the weathering of hosted minerals, resulting in the formation of new minerals, adsorption by organic and inorganic ligands, minerals, surface precipitation, and redox reactions [15–19]. It is also known that the sorting process accounts for significant REE accumulation, and human activity also contributes to the release of REEs in the environment [20].

Source rocks are the main provider of REEs in river sediments. REE content in the fluvial sediments of the LZR is mainly derived from the upper part of the drainage basin where the source area is characterized by a variety of igneous, metamorphic, and sedimentary rocks [21–23].

Due to the unique chemical properties of REEs, they are widely used in geochemical studies to identify weathering processes in river basins, tectonic setting, and provenance. The main objective of this study is to investigate the origin and tectonic setting of LZRB sediments by analyzing the comprehensive REEs datasets generated from detailed study of sediment samples collected from both the primary river and the inter-sub-basin regions. To the best of our knowledge, no previous study on the REE composition of LZRB sediments has been published. Here, for the first time, we present original data on the occurrence of REEs in LZRB sediments that have a bearing on its provenance and tectonic setting.

## 2. Methodology

### 2.1. Study Area

The LZR is one of the largest tributaries supplying water to the Tigris River, with the main portion of the drainage basin located in northeast Iraq and a small portion in northwest Iran [24]. The two largest permanent drainage basins in the LZRB are the Nirawan and Hami Qeshan sub-basins with areas of about 3074 km$^2$ and 4422 km$^2$, respectively, which join together to form the main course of the LZR. The LZRB has a length of approximately 374 km and covers a catchment area of approximately 20,000 km$^2$ [25]. The LZR, in its NE–SW trend, passes through many structures forming water gaps [26]. There are many large cities and towns like Penjween, QalaDiza, Raniyah, Koisanjaq, Dibis, and Altun Kupri inside Iraq, and Baneh, Sardasht, and Piranshar inside Iran, distributed within the basin in the upper, middle, and lower parts of the river (Figure 1).

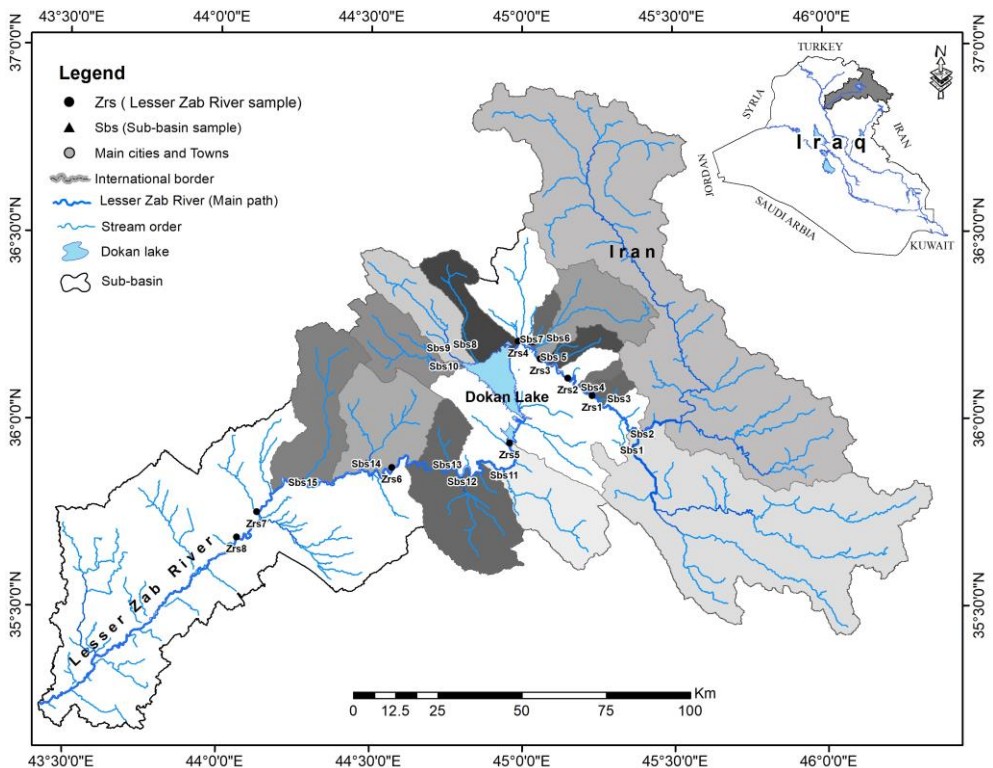

**Figure 1.** Location map showing the LZRB, LZR, active sub-basins, and the sampling sites, modified after [25].

The LZRB is part of the Zagros Orogenic Belt, which extends in an NW–SE direction from the East Anatolian Fault in southeastern Turkey through Iraq to the Oman Line in southern Iran and is characterized by complex tectonic and lithostratigraphy [27]. The belt marks the collision between the Arabian and Iranian plates resulting from the Late Cretaceous and Cenozoic convergence, when the intervening Neo-Tethys Ocean underwent a succession of subduction, obduction, and collision stages [28–31]. There are several rock types and sediments exposed in the basin from Precambrian to Quaternary [32–35]. The Iraqi part of the basin includes many types of ophiolite igneous complexes, pillow lavas, and sedimentary deposits [36–38]. Many studies have documented different types of igneous rocks, such as basalt, gabbro, syenite, metadiabases, diorite, peridotite, serpentinites, nepheline syenite, granitoid-gabbro pegmatites, and others; most of these rocks are affected by various degrees of deformation [39,40].

The Precambrian–Early Cambrian rocks represented by Soltanieh Dolomite, Barut, and Lalun formations occur in the black shale in the northeastern part of the study area inside Iranian territory and consist mainly of massive dolomite, dolomitic limestone, limestones,

and sandstone shelly pebbles [41,42]. In the upstream basin area, igneous and metamorphic rocks are exposed. The sedimentary rocks exposed in the middle and upper parts of the basin are a mix of carbonate and clastic rocks, whereas clastic sediments predominate in the lower part [43]. Figure 2 shows the geological map and various rock formations occurring in the LZR catchment.

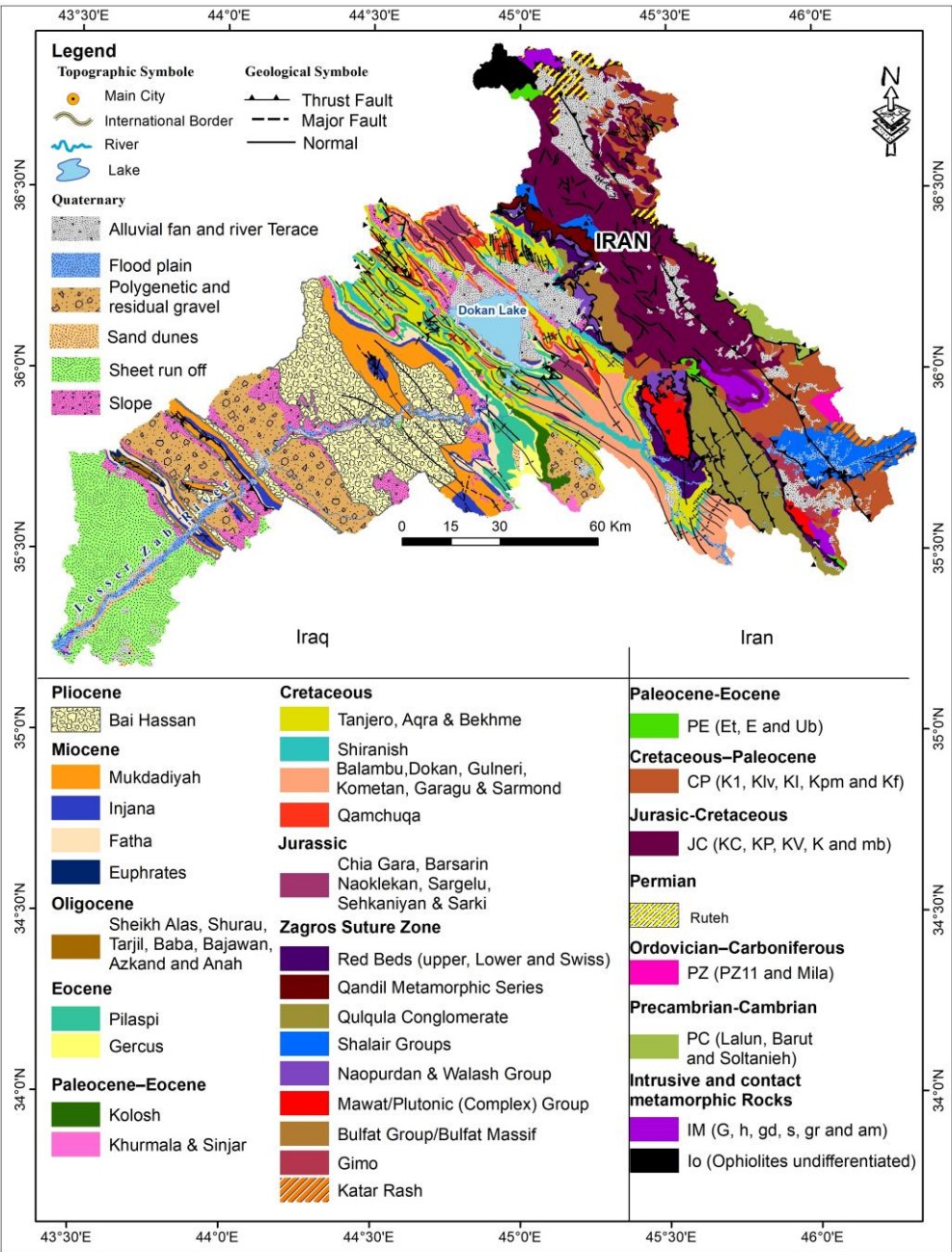

**Figure 2.** Generalized geologic map showing the main rock types (igneous, metamorphic, and sedimentary) in the LZR and its sub-basin catchment areas, modified after [33–36,44]. For more details about the lithology and age of each unit/formation, please see Appendix A.

Most of the geological units within the LZR basin comprise sedimentary rocks, dominated by limestones, dolomites, and marls; some clastic rocks are also present. The igneous rocks in the LZR basin are represented by Shalair, Mawat, Katar Rash, Intrusive Complex, Walash, and IM groups, while the metamorphic rocks include Gimo, Qandil, and CP groups [32–35,45].

### 2.2. Sampling and Analytical Methods

A total of 23 surface sediment samples were collected from LZRB: eight samples along the main course of the LZR, and 15 samples from sub-basins during the rainy season (Figure 1). The locations of samples were pre-selected based on preliminary survey results using morphometric analysis and fieldwork. They were chosen to represent most parts of the drainage basin. Due to anthropogenic activity, such as urban and agricultural developments, sampling sites were selected outside of the boundaries of the main cities and towns. All samples were taken from sediments in the main course of the perennial and intermittent rivers and some from locations underwater with low water levels. Approximately 2 kg of sediment was collected and stored in clean polyethylene bags in the field. All sediment samples were air-dried at ambient temperature before sieving. Sediment samples were sieved through a 2 mm sieve to remove coarse debris and gravel [14] to make the sediment more representative and homogeneous. 100 g of each sample was ground in an agate mortar to <0.045 mm in the laboratory of GEOSURV-Iraq to achieve a homogenized powdered form. A weight of 0.1 g fractions of powdered sediment was digested under the microwave conditions specified in the laboratory of TU Bergakademie Freiberg, Germany, before chemical analysis. 100 μL of internal standard (65% $HNO_3$ + 5 ppm Ge + 1 ppm Rh + 1 ppm Re) was added to the digested solution and volumetric flasks were filled to 10 mL using double-distilled water and analyzed using ICPMS. There are several studies in the LZRB dealing with trace elements, such as [44], therefore, we did not consider the whole suite of elements in this study but selected specific elements to support our objectives. Chemical analysis of REE concentrations was carried out by using standard methods and quality assurance and control (QA/QC) protocols. Triplicate samples were simultaneously prepared and analyzed using the same procedure to assess contamination and precision in the samples. The sampling procedure and analytical method used are believed to accurately represent REE compositions in the sediments of the LZR and its tributaries. One sample was split into three portions (triplicate split) in order to calculate the precision of geochemical analysis results and validate the laboratory tests. Furthermore, the accuracy of the analysis was checked using international reference standards. Additionally, we added three elements, germanium (Ge), rhenium (Re), and rhodium (Rh) to the digestion solution as an internal standard to monitor the efficiency of the equipment. To determine REE accumulation levels in the river sediments of the LZRB, we calculated the background concentration of europium ($^\delta$Eu) from Sm and Gd from Equation (1), while the cerium ($^\delta$Ce) anomalies background concentration was derived by interpolating between the normalized values of La and Pr (Equation (2)) [46]:

$$^\delta Eu = EuN/\sqrt{(SmN \cdot GdN)} \tag{1}$$

$$^\delta Ce = CeN/\sqrt{(LaN \cdot PrN)} \tag{2}$$

Subscript N indicates the normalized abundance with chondrite.

The ratio between LREEs (La–Sm) and HREEs (Gd–Lu) was obtained through calculation. The fractionation parameters between LREE and HREE concentrations were quantified by correlating between the $(La/Lu)_{UCC}$, $(Gd/Lu)_{UCC}$, $(La/Yb)_{UCC}$, $(Gd/Yb)_{UCC}$, and $(La/Sm)_{UCC}$ with the subscript UCC denoting normalized concentration. The ratios of $(La/Lu)_{UCC}$, and $(La/Yb)_{UCC}$ indicate the enrichment of LREEs relative to the HREEs, the ratio of $(La/Sm)_{UCC}$ refers to the fractionation of the LREE, and the ratios of the $(Gd/Yb)_{UCC}$ and $(Gd/Lu)_{UCC}$ refer to the fractionation of the HREEs [47,48].

### 3. Results

The concentrations of REEs in the analyzed samples are given in Table 1. The total REE (designated SREE) concentrations of LZR sediments ranged from 24.74 μg/g to 146.12 μg/g with a mean value of 76.12 μg/g, and for sub-basin sediments from 25.15 μg/g to 169.73 μg/g, with a mean value of 63.81 μg/g (Table 2). The standard deviations (SD) for LZR and sub-basins were 41.33 μg/g and 39.15 μg/g, respectively. The highest SREE

concentration for LZR sediments occured at the Zrs4 site with a value of 146.12 µg/g, and for the sub-basin at Sbs2 with a value of 169.73 µg/g (Table 2). REEs of the LZR and its sub-basin sediments were considerably enriched in LREEs relative to HREEs; enrichment of the LREEs from La to Eu was higher than HREEs from Gd to Lu. The ratio of LREEs/HREEs displays noticeable variation between samples, from 7.15 to 12.37 with a mean value of 9.68 and standard deviation ±1.76 for the LZR sediments and from 5.95 to 12 with a mean value of 8.77 and standard deviation ±1.72 for sub-basin sediments (Table 2).

**Table 1.** REE concentrations of surface sediments from the LZR and its sub-basins (µg/g).

| | | **Sub-Basin Samples** | | | | | | | | | | | | | |
| | | **LREEs** | | | | | | **HREEs** | | | | | | | |
| | S.ID. | La | Ce | Pr | Nd | Sm | Eu | Gd | Tb | Dy | Ho | Er | Tm | Yb | Lu |
| Sub-basin samples | Sbs1 | 20 | 40.1 | 4.03 | 15.47 | 3.18 | 0.71 | 2.93 | 0.42 | 2.21 | 0.43 | 1.17 | 0.15 | 0.90 | 0.12 |
| | Sbs2 | 37.87 | 77.07 | 7.44 | 27.87 | 5.42 | 1.01 | 4.76 | 0.66 | 3.4 | 0.65 | 1.78 | 0.23 | 1.39 | 0.19 |
| | Sbs3 | 9.24 | 18.55 | 2.35 | 9.60 | 2.11 | 0.55 | 2.12 | 0.32 | 1.75 | 0.35 | 0.97 | 0.13 | 0.78 | 0.11 |
| | Sbs4 | 8.02 | 15.58 | 1.96 | 7.89 | 1.75 | 0.45 | 1.79 | 0.27 | 1.49 | 0.30 | 0.82 | 0.11 | 0.65 | 0.09 |
| | Sbs5 | 24.74 | 50.21 | 4.94 | 18.91 | 3.82 | 0.84 | 3.47 | 0.48 | 2.54 | 0.49 | 1.32 | 0.17 | 1.04 | 0.14 |
| | Sbs6 | 14.47 | 34.84 | 3.50 | 13.39 | 2.64 | 0.61 | 2.42 | 0.34 | 1.76 | 0.33 | 0.88 | 0.11 | 0.65 | 0.09 |
| | Sbs7 | 13.48 | 24.58 | 3.08 | 11.90 | 2.44 | 0.57 | 2.35 | 0.33 | 1.73 | 0.34 | 0.92 | 0.12 | 0.70 | 0.10 |
| | Sbs8 | 11.05 | 21.75 | 2.52 | 9.61 | 1.93 | 0.43 | 1.79 | 0.24 | 1.22 | 0.23 | 0.61 | 0.08 | 0.47 | 0.07 |
| | Sbs9 | 7.92 | 16.07 | 1.92 | 7.50 | 1.57 | 0.36 | 1.49 | 0.21 | 1.10 | 0.21 | 0.58 | 0.07 | 0.45 | 0.06 |
| | Sbs10 | 5.15 | 9.14 | 1.17 | 4.71 | 1.05 | 0.30 | 1.11 | 0.17 | 0.99 | 0.20 | 0.56 | 0.07 | 0.45 | 0.06 |
| | Sbs11 | 6.95 | 12.91 | 1.64 | 6.48 | 1.35 | 0.34 | 1.32 | 0.19 | 1.01 | 0.20 | 0.53 | 0.07 | 0.42 | 0.06 |
| | Sbs12 | 12.91 | 31.87 | 3.23 | 12.64 | 2.66 | 0.63 | 2.44 | 0.34 | 1.75 | 0.33 | 0.85 | 0.11 | 0.65 | 0.09 |
| | Sbs13 | 10.67 | 21.18 | 2.61 | 10.19 | 2.18 | 0.53 | 1.99 | 0.27 | 1.37 | 0.25 | 0.65 | 0.08 | 0.49 | 0.07 |
| | Sbs14 | 6.81 | 12.89 | 1.66 | 6.58 | 1.43 | 0.36 | 1.34 | 0.19 | 0.96 | 0.18 | 0.47 | 0.06 | 0.36 | 0.05 |
| | Sbs15 | 9.46 | 18.78 | 2.33 | 9.23 | 1.98 | 0.48 | 1.86 | 0.26 | 1.33 | 0.25 | 0.66 | 0.08 | 0.51 | 0.07 |
| | Min | 5.15 | 9.14 | 1.17 | 4.71 | 1.05 | 0.30 | 1.11 | 0.17 | 0.96 | 0.18 | 0.47 | 0.06 | 0.36 | 0.05 |
| | Max | 37.87 | 77.07 | 7.44 | 27.87 | 5.42 | 1.01 | 4.76 | 0.66 | 3.40 | 0.65 | 1.78 | 0.23 | 1.39 | 0.19 |
| | Mean | 13.25 | 27.04 | 2.96 | 11.47 | 2.37 | 0.55 | 2.21 | 0.31 | 1.64 | 0.32 | 0.85 | 0.11 | 0.66 | 0.09 |
| | SD | 8.58 | 17.88 | 1.59 | 5.87 | 1.11 | 0.20 | 0.95 | 0.13 | 0.67 | 0.13 | 0.35 | 0.05 | 0.28 | 0.04 |
| | CV% | 64.73 | 66.13 | 53.69 | 51.18 | 47.07 | 36.02 | 42.83 | 41.39 | 40.63 | 40.49 | 41.45 | 42.24 | 41.88 | 39.84 |
| LZR samples | Zrs1 | 22.39 | 44.88 | 4.33 | 16.3 | 3.16 | 0.62 | 2.84 | 0.40 | 2.26 | 0.46 | 1.33 | 0.18 | 1.13 | 0.16 |
| | Zrs2 | 10.99 | 22.07 | 2.61 | 9.96 | 1.99 | 0.42 | 1.83 | 0.26 | 1.37 | 0.27 | 0.74 | 0.1 | 0.6 | 0.08 |
| | Zrs3 | 26.8 | 53.93 | 5.18 | 19.58 | 3.81 | 0.75 | 3.40 | 0.46 | 2.33 | 0.44 | 1.15 | 0.14 | 0.87 | 0.11 |
| | Zrs4 | 32.81 | 65.83 | 6.32 | 23.96 | 4.73 | 1 | 4.29 | 0.59 | 3.03 | 0.57 | 1.51 | 0.19 | 1.13 | 0.15 |
| | Zrs5 | 5.19 | 9.55 | 1.16 | 4.58 | 0.97 | 0.25 | 1.00 | 0.15 | 0.81 | 0.16 | 0.45 | 0.06 | 0.35 | 0.05 |
| | Zrs6 | 11.55 | 23.06 | 2.82 | 10.95 | 2.27 | 0.53 | 2.04 | 0.28 | 1.39 | 0.26 | 0.69 | 0.09 | 0.52 | 0.07 |
| | Zrs7 | 9.24 | 18.21 | 2.27 | 8.96 | 1.92 | 0.46 | 1.80 | 0.25 | 1.30 | 0.24 | 0.65 | 0.08 | 0.50 | 0.07 |
| | Zrs8 | 12.70 | 25.01 | 3.12 | 12.26 | 2.64 | 0.63 | 2.46 | 0.34 | 1.76 | 0.33 | 0.87 | 0.11 | 0.66 | 0.09 |
| | Min | 5.19 | 9.55 | 1.16 | 4.58 | 0.97 | 0.25 | 1 | 0.15 | 0.81 | 0.16 | 0.45 | 0.06 | 0.35 | 0.05 |
| | Max | 32.81 | 65.83 | 6.32 | 23.96 | 4.73 | 1.00 | 4.29 | 0.59 | 3.03 | 0.57 | 1.51 | 0.19 | 1.13 | 0.16 |
| | Mean | 16.46 | 32.82 | 3.48 | 13.32 | 2.69 | 0.58 | 2.46 | 0.34 | 1.78 | 0.34 | 0.92 | 0.12 | 0.72 | 0.1 |
| | SD | 9.69 | 19.67 | 1.69 | 6.26 | 1.19 | 0.23 | 1.04 | 0.14 | 0.72 | 0.14 | 0.37 | 0.05 | 0.29 | 0.04 |
| | CV% | 58.86 | 59.93 | 48.48 | 47.03 | 44.19 | 38.85 | 42.14 | 40.88 | 40.19 | 39.80 | 40.26 | 40.97 | 40.60 | 39.05 |
| | Chondrite | 0.37 | 0.96 | 0.14 | 0.71 | 0.23 | 0.09 | 0.31 | 0.06 | 0.38 | 0.09 | 0.25 | 0.04 | 0.25 | 0.04 |
| | NASC | 32 | 73 | 7.9 | 33 | 5.7 | 1.24 | 5.2 | 0.85 | 5.8 | 1.0 | 3.4 | 0.5 | 3.1 | 0.48 |
| | UCC | 30 | 64 | 7.10 | 26 | 4.5 | 0.88 | 3.8 | 0.64 | 3.5 | 0.8 | 2.3 | 0.33 | 2.20 | 0.32 |
| | BCC | 16.00 | 33 | 3.9 | 16 | 3.5 | 1.1 | 3.3 | 0.6 | 3.7 | 0.78 | 2.2 | 0.32 | 2.2 | 0.3 |

Where NASC is North American Shale Composite, UCC is Upper Continental Crust, and BCC is Bulk Continental Crust.

In general, the LREEs/HREEs of LZR and sub-basin tributaries sediments ratio is similar to the ratio of many rivers around the world [3,47]. The mean concentrations of REEs in the LZR and sub-basins sediments are found to be in the decreasing order of Ce > La > Nd > Pr > Sm > Gd > Dy > Er > Yb > Eu > Ho > Tb > Tm > Lu. Mean values of REE concentrations in LZR and its sub-basin sediments, compared to North American Shale Composite (NASC), Upper Continental Crust (UCC), and Bulk Continental Crust (BCC), are plotted and depicted in Figure 3. The sediment of the LZR and its sub-basin tributaries shows the same patterns as compared to reference values of NASC, UCC and BCC. All values of REEs from LZRB sediments show slightly lower concentrations than all reference values, while LREE values are closer to BCC reference values.

**Table 2.** Sum of REEs, LREEs, HREEs, LR/HR, and fractionation ratio in the fluvial sediments of LZR and sub-basin tributaries.

| | S.ID. | ∑REE | ∑LREE | ∑HREE | ∑LREEs/∑HREEs | $^{\delta}$EU | $^{\delta}$Ce | La/Lu | La/Sm | La/Yb | Gd/Yb | Gd/Lu |
|---|---|---|---|---|---|---|---|---|---|---|---|---|
| Sub-basin samples | Sbs1 | 91.82 | 83.49 | 8.33 | 10.02 | 0.71 | 1.05 | 1.81 | 0.94 | 1.63 | 1.89 | 2.09 |
| | Sbs2 | 169.73 | 156.68 | 13.05 | 12 | 0.61 | 1.08 | 2.18 | 1.05 | 2.00 | 1.99 | 2.16 |
| | Sbs3 | 48.93 | 42.41 | 6.52 | 6.51 | 0.8 | 0.93 | 0.89 | 0.66 | 0.87 | 1.58 | 1.6 |
| | Sbs4 | 41.16 | 35.64 | 5.52 | 6.46 | 0.77 | 0.92 | 0.91 | 0.69 | 0.91 | 1.60 | 1.61 |
| | Sbs5 | 113.12 | 103.46 | 9.66 | 10.71 | 0.71 | 1.06 | 1.85 | 0.97 | 1.74 | 1.93 | 2.05 |
| | Sbs6 | 76.02 | 69.45 | 6.57 | 10.57 | 0.73 | 1.15 | 1.80 | 0.82 | 1.64 | 2.17 | 2.38 |
| | Sbs7 | 62.63 | 56.04 | 6.58 | 8.51 | 0.73 | 0.89 | 1.45 | 0.83 | 1.40 | 1.93 | 1.99 |
| | Sbs8 | 52.01 | 47.29 | 4.72 | 10.02 | 0.70 | 0.97 | 1.76 | 0.86 | 1.71 | 2.18 | 2.25 |
| | Sbs9 | 39.52 | 35.35 | 4.17 | 8.47 | 0.73 | 0.97 | 1.33 | 0.75 | 1.29 | 1.91 | 1.97 |
| | Sbs10 | 25.15 | 21.53 | 3.62 | 5.95 | 0.86 | 0.87 | 0.85 | 0.74 | 0.84 | 1.43 | 1.45 |
| | Sbs11 | 33.47 | 29.67 | 3.8 | 7.81 | 0.78 | 0.9 | 1.24 | 0.77 | 1.22 | 1.83 | 1.86 |
| | Sbs12 | 70.49 | 63.93 | 6.56 | 9.75 | 0.75 | 1.16 | 1.55 | 0.73 | 1.46 | 2.19 | 2.32 |
| | Sbs13 | 52.55 | 47.37 | 5.18 | 9.15 | 0.77 | 0.94 | 1.68 | 0.73 | 1.59 | 2.35 | 2.48 |
| | Sbs14 | 33.33 | 29.72 | 3.61 | 8.24 | 0.8 | 0.9 | 1.45 | 0.72 | 1.38 | 2.14 | 2.25 |
| | Sbs15 | 47.28 | 42.27 | 5.01 | 8.43 | 0.77 | 0.94 | 1.43 | 0.72 | 1.37 | 2.13 | 2.22 |
| | Min | 25.15 | 21.53 | 3.61 | 5.95 | 0.61 | 0.87 | 0.85 | 0.66 | 0.84 | 1.43 | 1.45 |
| | Max | 169.73 | 156.68 | 13.05 | 12.00 | 0.86 | 1.16 | 2.18 | 1.05 | 2 | 2.35 | 2.48 |
| | Mean | 63.81 | 57.62 | 6.19 | 8.84 | 0.75 | 0.98 | 1.48 | 0.80 | 1.4 | 1.95 | 2.05 |
| | SD | 37.64 | 35.14 | 2.57 | 1.72 | 0.06 | 0.09 | 0.39 | 0.11 | 0.34 | 0.26 | 0.3 |
| LZR samples | Zrs1 | 100.46 | 91.68 | 8.78 | 10.44 | 0.63 | 1.07 | 1.52 | 1.06 | 1.45 | 1.46 | 1.52 |
| | Zrs2 | 53.28 | 48.04 | 5.23 | 9.18 | 0.67 | 0.97 | 1.41 | 0.83 | 1.35 | 1.78 | 1.85 |
| | Zrs3 | 118.96 | 110.06 | 8.90 | 12.37 | 0.64 | 1.07 | 2.50 | 1.05 | 2.27 | 2.27 | 2.50 |
| | Zrs4 | 146.12 | 134.66 | 11.46 | 11.75 | 0.68 | 1.07 | 2.33 | 1.04 | 2.13 | 2.20 | 2.41 |
| | Zrs5 | 24.74 | 21.70 | 3.04 | 7.15 | 0.79 | 0.91 | 1.09 | 0.80 | 1.08 | 1.65 | 1.66 |
| | Zrs6 | 56.51 | 51.18 | 5.33 | 9.60 | 0.75 | 0.95 | 1.69 | 0.76 | 1.62 | 2.26 | 2.35 |
| | Zrs7 | 45.95 | 41.06 | 4.89 | 8.39 | 0.76 | 0.93 | 1.42 | 0.72 | 1.36 | 2.09 | 2.19 |
| | Zrs8 | 62.98 | 56.37 | 6.61 | 8.53 | 0.76 | 0.93 | 1.48 | 0.72 | 1.41 | 2.15 | 2.26 |
| | Min | 24.74 | 21.70 | 3.04 | 7.15 | 0.63 | 0.91 | 1.09 | 0.72 | 1.08 | 1.46 | 1.52 |
| | Max | 146.12 | 134.66 | 11.46 | 12.37 | 0.79 | 1.07 | 2.50 | 1.06 | 2.27 | 2.27 | 2.50 |
| | Mean | 76.12 | 69.34 | 6.78 | 9.68 | 0.71 | 0.99 | 1.68 | 0.87 | 1.58 | 1.98 | 2.09 |
| | SD | 41.33 | 38.63 | 2.74 | 1.76 | 0.06 | 0.07 | 0.49 | 0.15 | 0.41 | 0.31 | 0.37 |

La/Lu, La/Sm, La/Yb, Gd/Yb, and Gd/Lu normalized to the Upper Continental Crust.

The concentrations of REEs have followed the Oddo–Harkins rule, where the even-numbered REEs have more abundance than odd-numbered elements [49,50], as shown in Table 1. To avoid the effect of the Oddo–Harkins, to identify the fractionation patterns of the REEs, and to assess the enrichment or depletion of the REEs in the sediments, concentrations of REEs were normalized to the reference values. The relative abundance of REEs in sediments of the LZR and its sub-basin tributaries was normalized to UCC, NASC, BCC, and chondrite (Figure 3a–h), which are considered the most frequently used in data normalization and result interpretation, calculated by dividing the concentration of the REE abundances by the REE concentration of the reference values. This approach allowed a reliable estimate of the overall composition of REEs.

The values of $^{\delta}$Eu range from 0.63 to 0.79 with a mean value of 0.71 for the LZR sediments and from 0.61 to 0.86 with a mean value of 0.75 for the sub-basin sediments. Therefore, sediment samples of the LZR and its sub-basins are characterized by a negative $^{\delta}$Eu anomaly. Values of the $^{\delta}$Ce anomaly vary from 0.91 to 1.07 with a mean value of 0.99 for the LZR sediments and from 0.87 to 1.16 with a mean value of 0.98 for sub-basin sediments. Most of the LZR samples, and about two-thirds of the sub-basin samples have a negative $^{\delta}$Ce anomaly.

The values of La/Lu, La/Sm, La/Yb, Gd/Yb, and Gd/Lu normalized to UCC in the LZR sediment samples range from 1.09 to 2.5, from 0.72 to 1.06, from 1.08 to 2.27, from 1.46 to 2.27, and from 1.52 to 2.5 with mean values of 1.68, 0.87, 1.58, 1.98, and 2.09, respectively. The values of the La/Lu, La/Sm, La/Yb, Gd/Yb, and Gd/Lu normalized to UCC in sub-basin sediment samples range from 0.85 to 2.18, from 0.66 to1.05, from 0.84 to 2, from 1.43 to 2.35, and from 1.45 to 2.48 with mean values of 1.48, 0.8, 1.4, 1.95 and 2.05, respectively (Table 2).

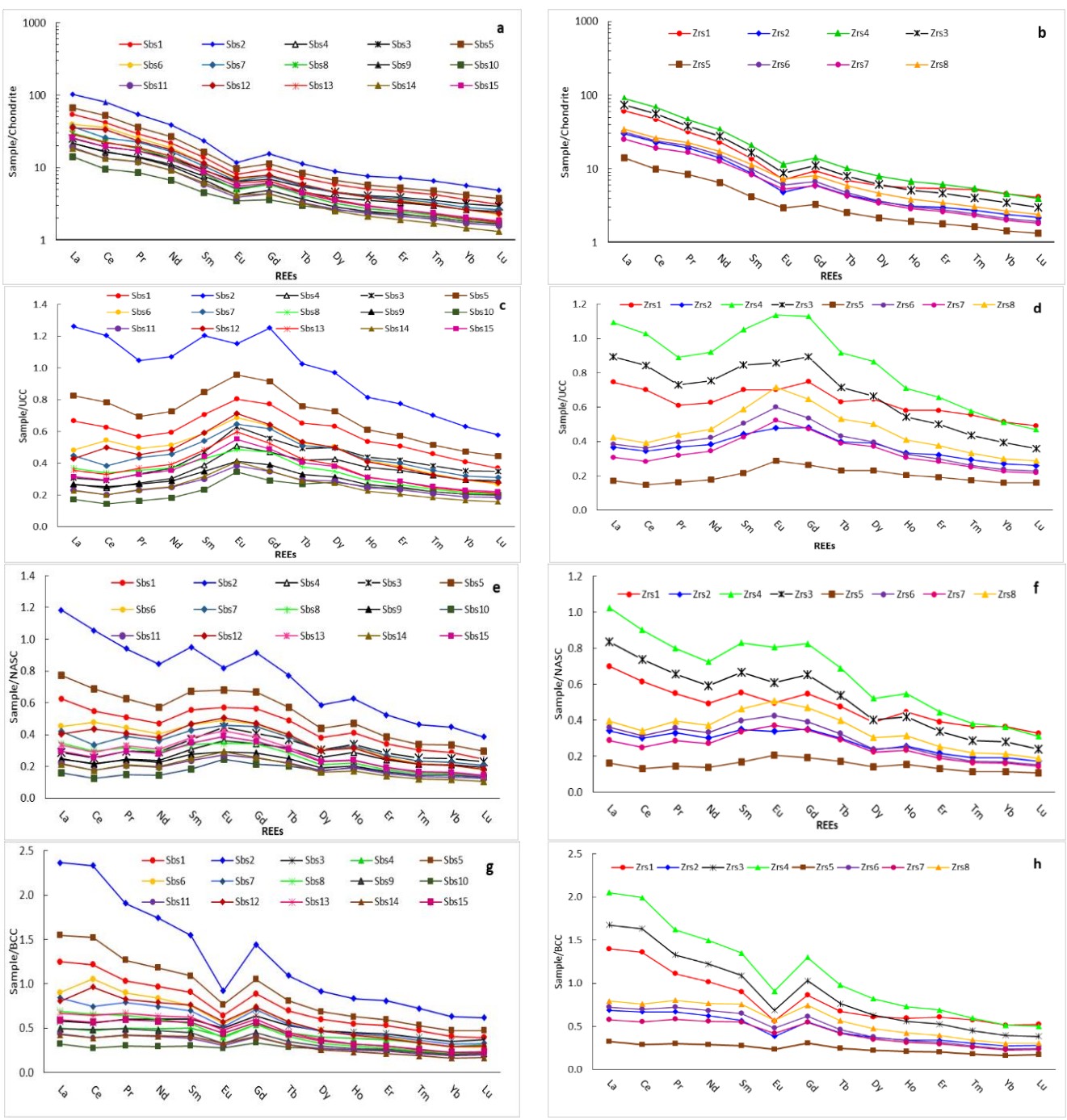

**Figure 3.** (**a**–**h**) show the UCC, NASC, BCC, and chondrite normalized REE patterns of the fluvial sediments of the LZR and its sub-basin tributaries.

## 4. Discussion

### 4.1. Normalization of REEs

Many researchers have used chondrite normalization to normalize the REEs in geological materials because the major components of the earth are similar to those of chondrite, and also to identify the origin of sediments [51]. The advantage of this method is that there is no considered fractionation between light and heavy REEs in chondrite [52]. Sediment samples of the LZR and sub-basin tributaries generally display enrichment patterns of LREEs normalized to chondrite and depletion in the HREEs with a general flat pattern for HREEs (Figure 3a). LREE enrichments are related to the dominance of plagioclase in

the source areas [1,53]. Most of the sub-basin samples exhibit similar REE concentration patterns and only three samples Sbs1, Sbs2, and Sbs5, show relatively high concentrations of LREEs (Figure 3a). Sub-basin samples Sbs1 and Sbs2 represent the largest permanent tributaries supplying water to the LZR and drain areas characterized by the exposure of different types of igneous, metamorphic, and sedimentary rocks. LZR samples show similar concentration patterns and only Zrs1, Zrs3, and Zrs 4, which were collected from the upper part of the main basin, have the highest concentration pattern of LREEs (Figure 3b). The uniform REE patterns of sub-basin Sbs1 and Sbs2, in addition to samples from upper part of LZR, may suggest all samples having the same paternal sources of REE and that all REE-bearing minerals are derived from igneous sources with negligible contributions from the clay-bearing sedimentary rocks in the area.

The noticeable variation in the LZR and sub-basin sediments from the upper part relative to the middle and lower parts reflects variation in source rocks. By comparing the REE pattern-normalized chondrite with the selected studies within the basin area from Kurdistan region, northern Iraqi territory, it can be inferred that it is compatible with the pattern of shale from Chia Gara Formation-normalized chondrite [54]. The pattern is incompatible with REE normalized-chondrite of the amphibolite rocks of the Penjween area in the northeastern part of the basin, which displays a regular pattern that denotes a paternal magma influenced by partial melting and fractional crystallization [55]. The sediments also have slight Eu depletion patterns and no Ce anomalies (Figure 3a,b). The lack of Ce anomalies among all the samples supports the conclusion that the oceanic-related sediments bearing REE are not involved as sources of the sediments in the studied river samples. In addition, although sediments from LZR and sub-basin tributaries have similar patterns, most samples from the upper-reach sediments have higher LREE concentrations than the middle- and lower-reach samples (Figure 3a,b).

The UCC-normalized REE pattern of sediments from the LZR and its sub-basin tributaries shows a general depletion pattern, having values of UCC-normalized REEs < 1. Only Zrs4 from the LZR samples is slightly enriched in some REEs (La, Ce, Sm, Eu, Gd), and Sbs2 from the sub-basin samples is enriched in REEs from (La to Tb) relative to UCC (Figure 3c,d). In general, UCC-normalized REE patterns show little or no Ce anomalies and noticeable Eu anomalies, which is attributed to lower Eu abundance in the upper crust, where Eu strongly partitions into plagioclase feldspar [56]. As mentioned earlier, there are three samples (Zrs1, Zrs3, and Zrs4) from the LZR and also three sub-basin samples (Sbs1, Sbs2, and Sbs5) that are relatively more enriched than other samples, which could imply different source rocks.

The NASC-normalized REE patterns (Figure 3e,f) of the LZR and sub-basin tributaries sediment obviously display a HREE-depleted pattern and have a convex pattern from (Nd-Dy) with a general trend similar to that of UCC-normalized pattern (Figure 3c,d). BCC-normalized patterns (Figure 3g,h) generally display a pattern similar to the chondrite-normalized pattern with negative Eu and positive Gd anomalies and more LREE enrichments relative to HREE for samples from the upper part of the basin relative to the lower part, while the samples of the middle and lower part have less fractionation.

Normalization of the results was challenging due to the complex lithology and tectonics of the study area. The exposed geological formations of the study area encompass a wide range from the Paleozoic era (541 million years before present) to the Holocene (present); in addition, the area is located within the complex tectonic setting of the Zagros Taurus range, which is characterized by frequent tectonic movements. Due to the lack of previous studies on the characterization of REEs in the region, we used several references from various sources to validate our results.

### 4.2. Fractionation Indices of REEs and $^{\delta}Eu$–$^{\delta}Ce$

Fractionation indices of REEs integrated with Eu and Ce anomalies can be considered the most important factor in identifying the source rock [1]. In the current study, as explained in Table 2, $^{\delta}$Eu values for the LZR samples and sub-basin sediments display low

values, corresponding to a negative Eu anomaly. This is also shown in Figure 4a. High LREE/HREE ratios corresponding to Eu anomalies may indicate the predominance of felsic igneous rocks [57,58]. The $^{\delta}$Ce values in samples of the LZR and sub-basin sediments show that some samples in the upper part of the main basin upstream of the Dokan Lake are slightly > 1 with a calculated mean value of about 1 for all samples, which indicate no noticeable anomaly (Table 2 and Figure 4a). $^{\delta}$Ce may indicate redistribution of REEs during weathering and as a consequence of fractionation.

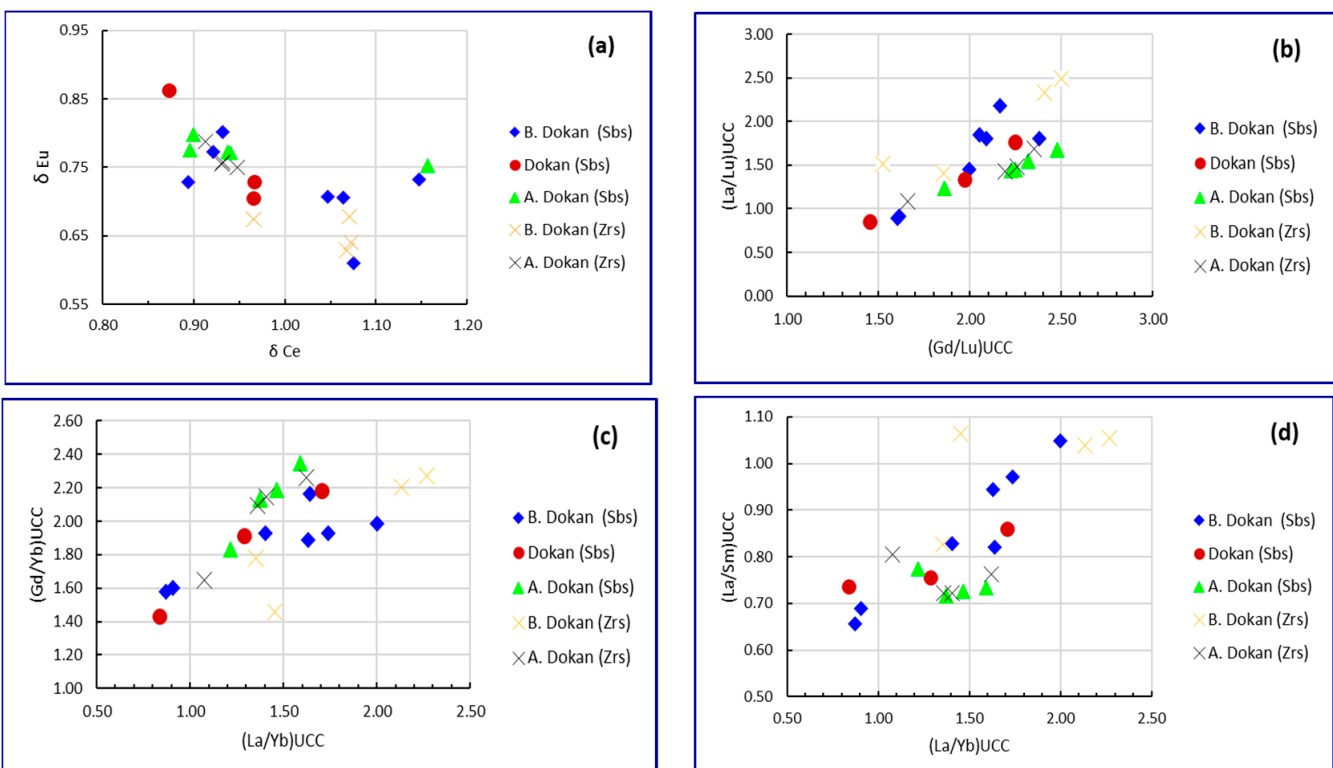

**Figure 4.** (**a**–**d**) show bivariate plots of $^{\delta}$Eu–$^{\delta}$Ce and the selected REE fractionation ratios for discriminating between the LZR and its sub-basin tributaries samples. Where [B. Dokan (Zrs)] refers to the sediment samples of the LZR upstream of the Dokan Lake, [A. Dokan (Zrs)] refers to the sediment samples of LZR downstream of the Dokan Lake, [B. Dokan (Sbs)] refers to the sediment samples of the sub-basin tributaries upstream of the Dokan Lake, [Dokan (Sbs)] refers to the sediment samples of the sub-basin tributaries discharged directly into the Dokan Lake, and [A. Dokan (Sbs)] refers to the sediment samples of the sub-basin tributaries downstream of the Dokan Lake.

Distinctive geochemical behavior of "$^{\delta}$Eu" relative to the other REEs is attributed to its substitution of Sr or Ca in feldspar under reducing conditions, where it may exist in the divalent state as within the mantle or lower crust [1,56]. Hence, $^{\delta}$Eu reflects earlier intra-crustal differentiation under a reducing igneous environment, where its enrichment is in the lower continental crust and deficiency in the upper continental crust [59]. Feldspar is considered the main factor controlling $^{\delta}$Eu during igneous processes and in felsic magma in particular, even though other minerals may also affect it.

The ratios of $(La/Lu)_{UCC}$ and $(La/Yb)_{UCC}$ for the LZR indicate that HREEs are depleted relative to LREEs. The UCC-normalized REE patterns of the LZR and its sub-basin fluvial sediments display significant fractionations of HRREs as marked by higher values of $(Gd/Lu)_{UCC}$ and $(Gd/Yb)_{UCC}$ relative to fractionations of LREEs as marked by $(La/Sm)_{UCC}$ (Figure 4b–d).

### 4.3. Comparing REEs in Sediments of LZR and Sub-Basins with Asian Rivers

Although the lithology and tectonic settings of the LZR and large rivers in Asian countries are different, their hydrology and hydrogeology are generally similar, and these control REE compositions, fractionations, and distribution pattern in alluvial sediments.

Due to the lack of published information on REE distribution and mobility resulting from weathering and alluvia transport for the study area, we elected to compare our results with rivers in Asian countries to gain a clear understanding of REE cycling in river sediments.

Values of $(La/sm)_{UCC}$, $(La/Yb)_{UCC}$, and $(Gd/Yb)_{UCC}$ in fluvial sediments of the LZR and its sub-basin sediments are similar to the values of the rivers in the Asian region. The other REE fractionation ratios of $(La/Lu)_{UCC}$ and $(Gd/Lu)_{UCC}$ are slightly higher than the ratios in rivers of the Asian region (Tables 2 and 3). The comparison of the LZR and its sub-basin sediment with selected large rivers from the Asian region shows that most of the studied samples have a relatively low value of $\sum$REE (Tables 2 and 3). There is only one sub-basin Sbs2 sample from the largest attribute [25] with an area of ~4422 km$^2$, which has a value of 169.73 µg/g higher than the Terengganu and Huanghe rivers, and the LZR sample designated Zrs4 has a value close to Huanghe river. The $\sum$REEs of the Asian river sediments are higher than those from the bulk samples of the LZR and sub-basin sediments, among which the LZR sample "Zrs4" and sub-basin sample "Sbs2" have the highest REE concentration, and LZR sample "Zrs5" and sub-basin "Sbs10" have the lowest. The low values of REE in the studied samples might be due to the predominance of limestone rocks in the drainage area, which is characterized by a low concentration of REEs. The $\sum$LREE/$\sum$HREE in the current study is similar to that of other rivers [3,47,60,61].

**Table 3.** The composition of REEs in selected rivers in the Asian region.

| River | Choshui (a) | Terengganu (b) | Huanghe (c) | Mekong (d) | Chao Phraya (e) | Yeongsan (f) | | | |
|---|---|---|---|---|---|---|---|---|---|
| Country | Taiwan | Malaysia | China | Asian Regions * | Thailand | Korea | Min | Max | Mean |
| $\sum$REE | 193.12 | 127.12 | 147.99 | 215.7 | 179.84 | 231.24 | 127.12 | 231.24 | 182.50 |
| $\sum$LREE | 173.67 | 118.39 | 132.76 | 193.5 | 157.98 | 188.52 | 118.39 | 193.50 | 160.80 |
| $\sum$HREE | 19.46 | 8.73 | 15.24 | 22.3 | 20.55 | 42.72 | 8.73 | 42.72 | 21.50 |
| $\sum$LREE/$\sum$HREE | 8.88 | 13.56 | 8.67 | 8.68 | 7.69 | 4.41 | 4.41 | 13.56 | 8.65 |
| $^{\delta}$Ce | 0.98 | 1.36 | 0.97 | 1 | 1.01 | 1.03 | 0.97 | 1.36 | 1.06 |
| $^{\delta}$Eu | 0.66 | 0.46 | 0.61 | 0.7 | 0.67 | 0.73 | 0.46 | 0.73 | 0.64 |
| $(La/Yb)UCC$ | 1.06 | 2.03 | 1.05 | 0.98 | 0.86 | 1.32 | 0.86 | 2.03 | 1.22 |
| $(La/Sm)UCC$ | 0.98 | 1.96 | 0.93 | 0.87 | 0.86 | 0.96 | 0.86 | 1.96 | 1.09 |
| $(Gd/Yb)UCC$ | 1.25 | 1.82 | 1.32 | 1.14 | 1.06 | 1.14 | 1.06 | 1.82 | 1.29 |
| $(Gd/Lu)UCC$ | 1.21 | 1.35 | 1.39 | 1.09 | 0.97 | 1.12 | 0.97 | 1.39 | 1.19 |
| $(La/Lu)UCC$ | 1.03 | 1.5 | 1.11 | 0.94 | 0.79 | 1.29 | 0.79 | 1.50 | 1.11 |

(a) [3] (Li et al., 2013), (b) [60] (Sultan & Shazili, 2009), (c), (d) and (e) [47] (Liu et al., 2019), (f) [61] (Xu et al., 2009), * China, Myanmar, Laos, Thailand, Cambodia, and Vietnam.

The mean enrichment ratios of $\Sigma$LREE over $\Sigma$HREE for the LZR and its sub-basin sediment are 9.68 and 8.84, respectively. They are the same as or within range of the Asian rivers, except for the Yeongsan River in Korea, with a value of 4.41. $^{\delta}$Ce and $^{\delta}$Eu values and other fractionation ratios are also within the same range relative to other rivers. All of the studied samples and the Asian rivers have a negative Eu anomaly ($^{\delta}$Eu < 1). Figure 5a shows that all Asian rivers have the same enrichment of LREEs relative to HREEs normalized to chondrite compared to the LZR and its sub-basin tributary pattern, and only the Terengganu River shows a different pattern compared to other rivers. The LREE enrichment chondrite-normalized pattern reflects the continental crust sediment pattern [3]. Furthermore, there is no $^{\delta}$Ce and slight $^{\delta}$Eu for studied sediments and Asian rivers, chondrite-normalized.

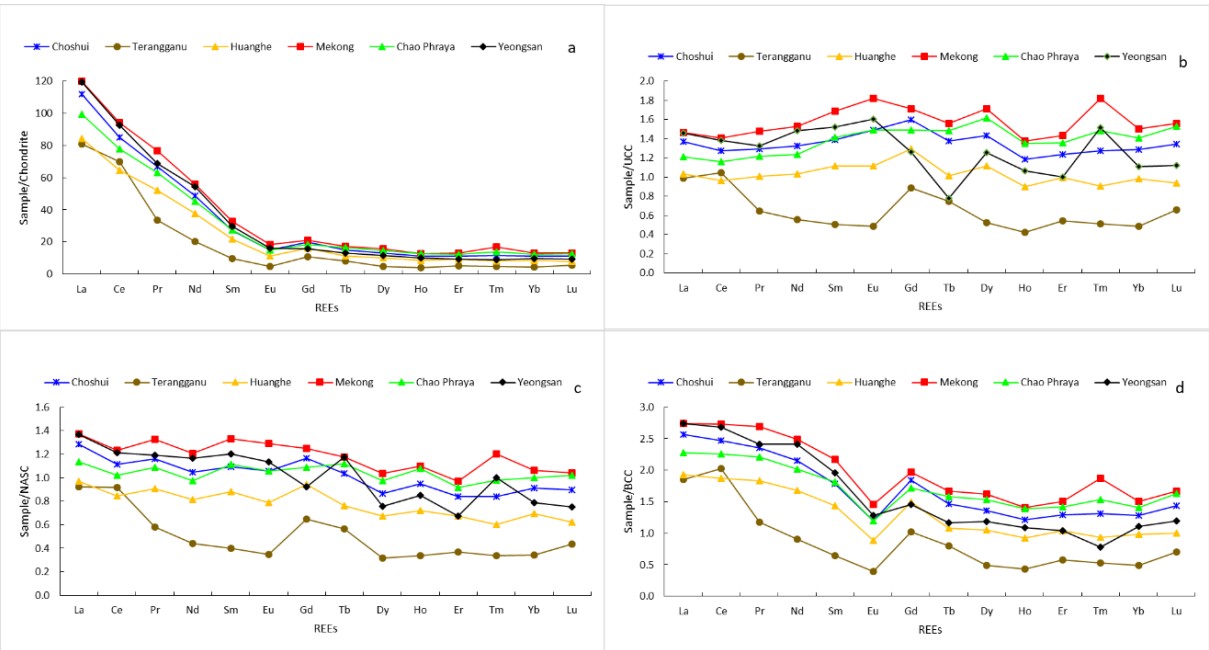

**Figure 5.** (**a**–**d**) showing chondrite-, UCC-, NASC-, and BCC-normalized REE patterns of selected Asian rivers.

There are different UCC- and NASC-normalized REE patterns between the Asian rivers and the LZR and its sub-basin sediments (Figure 5b,c), suggesting that the fluvial sediments of these rivers originate from various sources and are subjected to different geological conditions. Despite the different patterns between the Asian rivers and the LZR and its sub-basin sediments pattern, they as a whole have the same character of Eu anomaly and Gd enrichment.

### 4.4. Tectonic Setting and Provenance

The LZRB covers part of the major Zagros Orogenic Belt, which includes the Sanandaj–Sirjan from the northeast to the southwest, and the Zagros Fold and Thrust Belt within which occur the Suture, Imbricate, High Folded, and Low Folded zones inside Iraq [62,63]. Despite the source rocks being the major factor in controlling the composition of the sediment, other factors also play important roles, including grain size, climate, tectonic setting, hydraulic sorting adsorption on suspended particles, degree of chemical weathering, diagenesis, and metamorphism [1,64].

Rocks exposed on the LZRB catchment consist mainly of carbonate rocks, which are characterized by low REE contents. In the upper part of the main basin, Paleozoic and Mesozoic carbonate rocks dominate, with the presence of various types of intrusive and contact metamorphic rocks, while carbonate rocks and Quaternary clastic sediments dominate the middle and lower parts. These rocks in the LZR and its sub-basins are the source of sediments in the river and streams.

The provenance of REEs in the LZR and its sub-basin sediments, characterized by LREE enrichment and a relatively flat HREE pattern associated with a negative Eu anomaly, reflect the Upper Continental Crust. The REE normalized patterns and Eu anomaly can be utilized to identify sources of fluvial sediments and sedimentary rocks [1,56,65]. Also, mechanical weathering in LZR is more intense than chemical weathering compared to the Asian rivers due to the high altitude and semi-arid climate. However, higher SREE concentration in the sub-basin sample Sbs2 of the largest tributary supplying water to the LZR is related to the igneous and metamorphic rocks, which are widely exposed within this sub-basin and have relatively high REE concentration. Mechanical mixing of detritus

flux from sub-basin tributaries is probably the main factor controlling REE content along the main course of LZR.

Geochemical analysis is considered one of the important methods to discriminate the tectonic setting of sedimentary basins [66,67]. The mineralogy of river sediments reflects the source rocks; consequently, chemical composition has been widely utilized to recognize tectonic setting as well as provenance. The plot of Th versus Sc [68] (Figure 6f) shows that the sediment data scatter of Th/Sc is <1, with most samples that have Sc contents falling within the mafic signature, indicating a more mafic source. The mean value of the Th/Sc ratio is around the basalt value. Th/Sc ratios near 0.6 suggest a more mafic component. The Th–Sc diagram, further shows that sedimentary provenance in the LZR and sub-basin tributaries behaves mainly as mafic to intermediate provenance (Figure 6f).

The tectonic setting of the LZRB as a part of the Zagros Orogenic Belt means that it has experienced different tectonic conditions from rifting to subduction and final collision [27,29–31,38,69]. The Neo-Tethys Ocean evolved during the Permian period, when macroblocks rifted from Gondwana's northern margin and collided with the Eurasian continent in the Late Triassic, resulting in the closure of the Paleo-Tethys Ocean, followed by subduction of the Neo-Tethys Oceanic below the southern margin of the Eurasian continent, resulting in the closure of the Neo-Tethys Ocean because of collision and the formation of the Alps–Zagros–Himalaya Orogenic Belt [70].

REEs and some high-field strength elements are very useful in identifying source properties of sediments and clastic sedimentary rocks [1,71,72]. They are less reactive and undergo minor local variations or slight fractionation during transportation and deposition of sediments; therefore, they can effectively reflect the tectonic environment of sedimentary basins.

The ternary of the La-Th-Sc and La/Yb versus REE, Th/Co vs. La/Sc, La/Th vs. Th/Yb, and Co/Th versus La/Sc ratios plot offers a useful approach to discriminating the tectonic setting and identifying source-rock types [66,73–76]. Four distinctive tectonic settings are recognized on the ternary plots of the La-Th-Sc: the oceanic island arc, continental island arc, active continental, and passive continental margin sediments; the tectonic setting of the sediments is mainly the continental island arc (Figure 6a and Table 4).

LZR and its sub-basin tributary sediments on the La-Sc edge plot mainly closer to La, within the meta-basic source zone and the overlap area between the mixed and meta-basic sources and meta-basic sources of high silica content and only a few samples are within amphibolite sources of relatively low silica content. This may indicate that the andesitic unit within the Late Cretaceous Walash group [77] can be considered the main source of the sediments under investigation. The results indicate that the amphibolites have a relatively small contribution to the sediments relative to the granitic gneiss rocks or the metasediments derived from them. The lack of dominant amphibolite rock in the source region supports excluding amphibolite as a source of REEs in the sediments. Moreover, the enrichment of LREEs over HREEs in all studied samples excludes the dominance of garnet-bearing rocks in the source region. The cluster of samples with the island arc field, may suggest arc affinity sources of the studied sediments (Figure 6a). ΣREES versus La/Yb discrimination shows that the sediment samples are located mainly within the overlap zone of sedimentary rock, tholeiite, alkali basalts, and granite (Figure 6b). Most of the samples of the La/Sc and Th/Co ratios generally fall into the source area of felsic rocks (Figure 6c). The La/Th and Th/Yb ratios of all samples are plotted in Figure 6d. The ratios fall into the source area of the felsic and basic mixture and very few of the samples that tend towards mafic provenance also suggest dominantly felsic source rocks. The Co/Th versus La/Sc diagram indicates that the source rocks were mainly basalt and andesite (Figure 6e). The noticeable variability of provenances inferred from geochemical evidence points to inheritance from a mixture of lithotypes of various source rocks exposed in different tectonic zones in the main basin, particularly those outcropping in upstream areas with higher elevation, which become easily eroded and incorporated into the supplying sediments. This indicates that inputs of local lithological units significantly influence sediment composition.

The existence of enormous amounts of basaltic and andesitic rocks in the source region related to the Walash–Noupradan groups supports the outcomes of the above REE-discrimination diagrams.

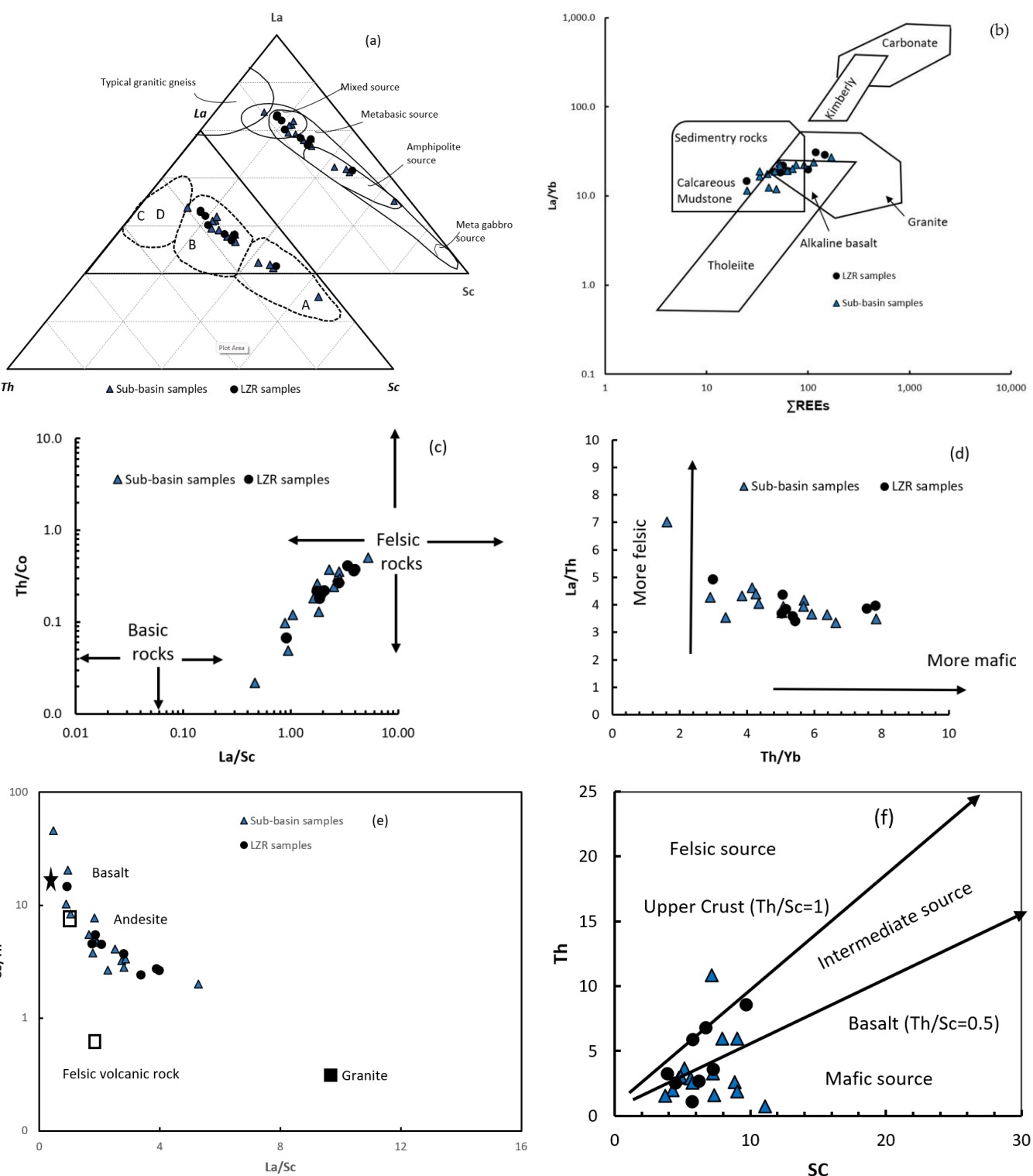

**Figure 6.** (**a**) La-Th-Sc plot showing various tectonic settings: A—oceanic island arc; B—continental island arc; C—active continental margin; D—passive margins [64,66,78]; (**b**) La/Yb versus REE [73]; (**c**) Th/Co versus La/Sc [75]; (**d**) La/Th versus Th/Yb [1]; (**e**) Co/Th versus La/Sc plot [76]; (**f**) Th versus Sc plot [68].

**Table 4.** Other trace-element data (μg/g) and fractionation indices for studied samples in the fluvial sediments of LZR and sub-basin tributaries.

| | S.ID. | Yb | Sc | Co | La | Th | La/Sc | Th/Co | Th/Yb | La/Th | Co/Th | Th/Sc |
|---|---|---|---|---|---|---|---|---|---|---|---|---|
| Sub-basin samples | Sbs1 | 0.90 | 7.96 | 24.43 | 20.00 | 5.97 | 2.51 | 0.24 | 6.64 | 3.35 | 4.09 | 0.75 |
| | Sbs2 | 1.39 | 7.18 | 21.66 | 37.87 | 10.87 | 5.28 | 0.50 | 7.83 | 3.49 | 1.99 | 1.51 |
| | Sbs3 | 0.78 | 8.83 | 21.73 | 9.24 | 2.61 | 1.05 | 0.12 | 3.36 | 3.54 | 8.32 | 0.30 |
| | Sbs4 | 0.65 | 9.02 | 19.24 | 8.02 | 1.88 | 0.89 | 0.10 | 2.90 | 4.26 | 10.23 | 0.21 |
| | Sbs5 | 1.04 | 9.03 | 19.16 | 24.74 | 5.95 | 2.74 | 0.31 | 5.70 | 4.16 | 3.22 | 0.66 |
| | Sbs6 | 0.65 | 5.15 | 10.35 | 14.47 | 3.67 | 2.81 | 0.35 | 5.68 | 3.94 | 2.82 | 0.71 |
| | Sbs7 | 0.70 | 4.72 | 9.72 | 13.48 | 2.92 | 2.86 | 0.30 | 4.15 | 4.62 | 3.33 | 0.62 |
| | Sbs8 | 0.47 | 4.86 | 8.08 | 11.05 | 3.03 | 2.28 | 0.38 | 6.39 | 3.65 | 2.66 | 0.62 |
| | Sbs9 | 0.45 | 4.27 | 10.10 | 7.92 | 1.96 | 1.86 | 0.19 | 4.36 | 4.04 | 5.16 | 0.46 |
| | Sbs10 | 0.45 | 11.11 | 33.52 | 5.15 | 0.73 | 0.46 | 0.02 | 1.63 | 7.02 | 45.69 | 0.07 |
| | Sbs11 | 0.42 | 7.35 | 32.98 | 6.95 | 1.61 | 0.95 | 0.05 | 3.85 | 4.32 | 20.48 | 0.22 |
| | Sbs12 | 0.65 | 7.26 | 12.43 | 12.91 | 3.28 | 1.78 | 0.26 | 5.08 | 3.93 | 3.79 | 0.45 |
| | Sbs13 | 0.49 | 5.58 | 13.62 | 10.67 | 2.91 | 1.91 | 0.21 | 5.92 | 3.66 | 4.67 | 0.52 |
| | Sbs14 | 0.36 | 3.71 | 11.95 | 6.81 | 1.55 | 1.84 | 0.13 | 4.27 | 4.39 | 7.70 | 0.42 |
| | Sbs15 | 0.51 | 5.76 | 13.95 | 9.46 | 2.54 | 1.64 | 0.18 | 5.03 | 3.72 | 5.49 | 0.44 |
| | Min | 0.36 | 3.71 | 8.08 | 5.15 | 0.73 | 0.46 | 0.02 | 1.63 | 3.35 | 1.99 | 0.07 |
| | Max | 1.39 | 11.11 | 33.52 | 37.87 | 10.87 | 5.28 | 0.50 | 7.83 | 7.02 | 45.69 | 1.51 |
| | Mean | 0.66 | 6.79 | 17.53 | 13.25 | 3.43 | 2.06 | 0.22 | 4.85 | 4.14 | 8.64 | 0.53 |
| | SD | 0.28 | 2.14 | 8.13 | 8.58 | 2.52 | 1.16 | 0.13 | 1.59 | 0.88 | 11.24 | 0.34 |
| LZR samples | Zrs1 | 1.13 | 5.79 | 16.12 | 22.39 | 5.84 | 3.87 | 0.36 | 5.17 | 3.84 | 2.76 | 1.01 |
| | Zrs2 | 0.60 | 3.93 | 12.04 | 10.99 | 3.24 | 2.80 | 0.27 | 5.44 | 3.40 | 3.72 | 0.82 |
| | Zrs3 | 0.87 | 6.76 | 18.01 | 26.80 | 6.77 | 3.96 | 0.38 | 7.82 | 3.96 | 2.66 | 1.00 |
| | Zrs4 | 1.13 | 9.73 | 20.67 | 32.81 | 8.53 | 3.37 | 0.41 | 7.57 | 3.85 | 2.42 | 0.88 |
| | Zrs5 | 0.35 | 5.75 | 15.53 | 5.19 | 1.06 | 0.90 | 0.07 | 2.99 | 4.91 | 14.69 | 0.18 |
| | Zrs6 | 0.52 | 6.24 | 14.58 | 11.55 | 2.64 | 1.85 | 0.18 | 5.07 | 4.37 | 5.51 | 0.42 |
| | Zrs7 | 0.50 | 4.52 | 11.37 | 9.24 | 2.51 | 2.05 | 0.22 | 5.04 | 3.68 | 4.53 | 0.56 |
| | Zrs8 | 0.66 | 7.29 | 16.28 | 12.70 | 3.55 | 1.74 | 0.22 | 5.36 | 3.58 | 4.59 | 0.49 |
| | Min | 0.35 | 3.93 | 11.37 | 5.19 | 1.06 | 0.90 | 0.07 | 2.99 | 3.40 | 2.42 | 0.18 |
| | Max | 1.13 | 9.73 | 20.67 | 32.81 | 8.53 | 3.96 | 0.41 | 7.82 | 4.91 | 14.69 | 1.01 |
| | Mean | 0.72 | 6.25 | 15.58 | 16.46 | 4.27 | 2.57 | 0.26 | 5.56 | 3.95 | 5.11 | 0.67 |
| | SD | 0.29 | 1.79 | 3.02 | 9.69 | 2.52 | 1.11 | 0.12 | 1.53 | 0.48 | 4.02 | 0.30 |

Despite some bivariate and ternary diagrams that could produce useful information, none of them can be completely satisfactory [79]. Therefore, the more precise discrimination of tectonic settings based on geochemical data requires applying several plots.

## 5. Conclusions

In the LZRB, Paleozoic calcareous sedimentary rocks with igneous and metamorphic rocks dominate the upper part, while carbonate, clastic, and Quaternary sediments dominate the middle and lower parts. All samples show enrichment of LREES relative to the HREE flat pattern for HREEs normalized to chondrite, and an Eu anomaly which correlates well with the UCC mean-value composition pattern.

The fluvial sediments of LZR and sub-basin tributaries display the same patterns normalized to chondrite, NASC, UCC, and BCC reference values. Studied samples usually show little variation in the relative rare-earth content of sediments except for a few samples. All values of REEs from LZRB sediments showed slightly lower concentrations than all reference values and LREEs are closer to the reference value of BCC. REE content in the fluvial sediment of LZR and sub-basin tributaries is lower than those of Asian rivers, due to the abundance of carbonate rocks within the main basins.

REE ratios indicate multisource rocks with a prevalence of felsic provenance. The La-Th-Sc plots suggest the tectonic setting environments of the LZRB sediments to be mainly of a continental island arc. Most sediment samples of LZRB are derived from elevated land, suggesting that physical weathering of bedrock controls the composition of REEs rather

than chemical weathering within the basin. The relatively high concentration in the LZR and sub-basin tributary sediments from the upper part of the basin, which is characterized by the exposure of a wide range of igneous and metamorphic rocks compared to the middle and lower parts, implies that bedrock composition is the primary controlling factor for REE composition of sediments. The above evidence also implies that a low abundance of REEs relative to UCC and NASC could be attributed to the dilution of quartz and carbonate minerals, where grain size plays an important role, because, in this study, we analyzed the <2 mm fraction. Hence, we can conclude that REEs' abundance in LZR and sub-basin sediments is controlled mainly by bedrock composition, type of weathering, and texture.

**Author Contributions:** Y.I.A.-S.: funding acquisition, conceptualization, resources, methodology, validation, formal analysis, visualization, writing—original draft. A.A.O.: supervision, conceptualization, resources, methodology, validation, writing—review and editing. Y.O.M.: conceptualization, methodology, writing—review and editing. S.S.A.: funding acquisition, resources, methodology, writing—review and editing. S.A.A.: conceptualization, methodology, writing—review and editing. V.L.: funding acquisition, resources, methodology, writing—review and editing. S.E.H.: conceptualization, methodology, writing—review and editing. All authors have read and agreed to the published version of the manuscript.

**Funding:** Provided by the Ministry of Higher Education and Scientific Research of the Iraqi Government under a twinning program with the TU Bergakademie Freiberg.

**Data Availability Statement:** The data presented in this study are available on request from the corresponding author.

**Acknowledgments:** The authors acknowledge the funding provided by the Ministry of Higher Education and Scientific Research of the Iraqi Government under a twinning program with the TU Bergakademie Freiberg. We are grateful to the Iraq Geological Survey for supporting us during fieldwork. We would like to thank Fouad S. Al-Kaabi and Rand M. Al-Saati for their valuable comments and suggestions.

**Conflicts of Interest:** The authors declare no conflict of interest.

## Appendix A

Brief description of the stratigraphy of LZRB within the Iranian part [21,22,32–35,42,45].

| Name | Age | Lithology of Iranian Part |
|------|-----|---------------------------|
| Soltanieh Dolomite Fn | Precambrian | Dolomite, with a shale intercalation in the lower part. |
| Barut Fn | Precambrian–Early Cambrian | Shales, with thin dolomites and limestones. |
| Lalun Fn | | Limestones and sandstones. |
| Mila Fn | Ordovician | Dolomites, limestones, marls, shales, and somewhat sandy beds. |
| Pz11 | Ordovician–Carboniferous | Crystallized limestone. |
| Ruteh Fn | Late Permian | Limestone. |
| KC, KP, KV, K, and Mb | Jurasic–Cretaceous | Mb: Marble; Kp: Homogenous phyllite; Kv: Green andesite and related tuffs; Kc: Conglomerate; and K: limestone, dolomite with subordinate shale. |
| K1, Klv, KI, Kpm, and Kf | Cretaceous–Paleocene | Kf: Low-grade metamorphism in general, Flysch-type facies with turbidites; Kpm: Low-grade metamorphism in general, mainly phyllite with minor limestone and volcanics; KI: Crystalized limestone and marble in parts affected by late Eocene thermic events; K11: Orbitolina, in parts, interbedded with slates or shales; Kiv: Andesitic volcanic and associated pyroclastic rocks, mainly lower cretaceous. |
| Et, E, and Ub | Paleocene–Eocene | E: Shale, sandy shale, sandstone with some fine limestone intercalations, andesitic to basaltic volcanic with pillow structures; Et: Andesitic pyroclastics, mainly crystal and lithic tuff; Ub: Ultrabasic rocks. |

| Name | Age | Lithology of Iranian Part |
|---|---|---|
| **Intrusive and contact metamorphic Rocks** | | |
| Gr, G, and gd | gr (Post-Cretaceous–Paleocene) G (Late Eocene–Early Oligocene) gd (Late Paleocene) | Intrusive rocks; gr: Granite; G: Gabbro to diorite with ultrabasic inclusion; gd: Biotite Granodiorite and its marginal varieties. |
| S, h, and am | Post-Cretaceous–Paleocene | S: Slate andalusite and schist; h: Pyroxene hornfels facies; and am: Amphibolite |
| Io | Post-Cretaceous–Paleocene | Ophiolites undifferentiated |
| **Lithostratigraphy of LZRB Iraqi part/Unstable Shelf** | | |
| Sarki | Early Liassic | Cherty dolomitic limestone with cherty shale and dolomite. |
| Sehkaniyan | Liassic | Lower unit: dolomites and dolomitic limestones with some solution breccia. Middle unit: fossiliferous limestone often dolomitized with some chert bands. Upper unit: Dolomites and dolomitic limestones, locally with chert. |
| Sargelu | Middle Jurassic (Bajocian–Bathonian) | Bituminous and dolomitic limestones, shaley limestone, and shales with chert and dolomitic marls. |
| Naokelekan | Late Jurassic | Lower unit: argillaceous bituminous limestone alternating with bituminous shale and fine-grained limestone. Middle unit: fossiliferous dolomitic limestone as "Mottled Beds". Upper unit: highly bituminous dolomite and limestone with beds of black shale. |
| Barsarin | Late Jurassic | Limestone and dolomitic limestone. |
| Chia Gara | Middle Tithonian–Berriasian | Limestone and calcareous shale. |
| Garagu | Late Berriasian–Hauterivian | Oolitic sandy limestones with marls and sandstones. |
| Lower Sarmord | Hauterivian–Berremian | Marls, with beds of argillaceous limestone. |
| Balambo | Valanginian–Middle Albian | Limestones, with beds of marl and shale. |
| Qamchuqa | Hauterivian–Albian | Limestones. |
| Dokan | Cenomanian | Oligosteginal limestone. |
| Gulneri | Lower Turonian | Black bituminous shale with glauconite and collophane in the lower part. |
| Kometan | Turonian | Globigerinal-oligosteginal limestone. |
| Bekhme | Late Campanian | Bituminous secondary dolomite. |
| Aqra | Maastrichtian | Limestone. |
| Shiranish | Late Campanian–Maastrichtian | Argillaceous limestones. |
| Tanjero | Late Campanian–Maastrichtian | Alternation of shale, claystone, sandstone, and siltstone, with limestone. |
| Kolosh | Early–Late Paleocene | Fine clastics, like sandstone, siltstone, and claystone. |
| Sinjar | Early Eocene | Fossiliferous limestone with occasional beds dolomitic limestone. |
| Khurmala | Lower Paleocene–Lower Eocene | Limestones and dolostones interfingering with limestones of Sinjar Formation. |
| Gercus | Early–Middle Eocene | Shales, mudstones, sandy and gritty marls, pebbly sandstones, and conglomerates. |
| Pila Spi | Middle–Late Eocene | Bituminous, chalky, and crystalline limestones. |
| Shurau | Early Oligocene | Coralline limestone. |
| Sheikh Alas | Oligocene | Porous, occasionally rubbly dolomitic, and recrystallized limestones. |
| Tarjil | Early Oligocene | Splintery limestone. |
| Bajawan | Late Oligocene | Reef miliolid limestones alternating with porous, dolomitized, reef limestones. |
| Baba | Middle Oligocene | Chalky limestone. |
| Anah | Late Oligocene | Brecciated recrystallized, detrital, and coralline limestones. |
| Azkand | Late Oligocene | Thick massive, dolomitic, and recrystallized, generally porous limestones. |
| Euphrates | Early Miocene | Shelly, chalky, and well-bedded recrystallized limestone, green marls, argillaceous sandstones, breccias, and conglomerates. |
| Fatha | Middle Miocene | Cyclic deposits of marl, limestone, gypsum, reddish brown claystone marls rather than green, with alternation of thick limestone |
| Injana | Late Miocene | Brown and gray sandstone interbedding with brown claystone and reddish-brown siltstones in cyclic nature. |
| Mukdadiya | Late Miocene | Alternation of claystone with cross-bedded sandstone, and brown and gray siltstone. |
| Bai Hassan | Late Miocene-Pliocene | Thick and coarse conglomerates alternating with thick brown claystones and thin sandstones. |

| Name | Age | Lithology of Iranian Part |
|---|---|---|
| **Lithostratigraphy of LZRB Iraqi part/Zagros Suture zone** | | |
| Qulqula Radiolarian | Barremian–Alpian | Thick bedded, oolitic, and detrital limestones, and thick beds of white chert, both interbedded with marly shale. |
| Qulqula Conglomerate | Albian–Cenomanian | Thick lenticular beds of conglomerates, composed of pebbles and small boulders of limestone, and to a lesser extent of chert. |
| Mawat group | Albian–Cenomanian | Pillow basalt, amygdaloidal basalt, spilite, and keratophyres, metamorphosed into greenschist facies and intruded by gabbro and ultrabasic rocks. |
| Gimo | Albian–Cenomanian | Massive and thick-bedded marble and calcschist interbedded with basaltic flows. |
| Qandil series | Cretaceous | Limestone, with some serpentinite intrusions. |
| Shalair series | Early–Late Cretaceous | Chlorite–sericite phyllite, in the lower part, interbedding with quartzite, and greywacke is common. |
| Katar Rash group | Late Cretaceous | Predominantly of calc-alkaline volcanics of andesite–rhyolite association. The most common rocks are andesites, dacite, and rhyolites. |
| Intrusive Complex | Early–Late Cretaceous | Intrusive complex of Bulfat massif (Late Cretaceous or younger) unit consists of igneous and metamorphic rocks only, amphibole diorite, olivine diorite, granodiorite, pegmatite syenite, and nepheline syenite. |
| Walash group | Late Cretaceous | Very thick basic volcanic sequence including conglomerate, lava flows, pillow lavas, and ashes with associated dykes. |
| Red Bed series | Paleocene–Miocene | Sequence of conglomerates and red and bluish-purple shale. |

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
