# Peer review of "Composition of Rare Earth Elements in Fluvial Sediments of the Lesser Zab River Basin, Northeastern Iraq: Implications for Tectonic Setting and Provenance"

_geosciences, doi:10.3390/geosciences13120373_

Round 1

Reviewer 1 Report

Comments and Suggestions for Authors

This study dealt with REE geochemistry in fluvial sediments of Lesser Zab River Basin. Based on the concentration and fractionation of REE and some other trace elements in sediments, implication for tectonic setting and provenance was discussed. I would like basically agree with the arguments presented by the authors. However, some interpretations still need to be revised, or even be changed.

1.     According to the title, authors may want to explore the implication of REE geochemistry for tectonic setting and provenance. However, one cannot see any information on this objective in abstract, which should be revised accordingly.

2.    Lines 65-66; Line 73-74. Related references should be added to support the review.

3.    The introduction is relatively simple. It should be further enriched with more previous studies related to the research topic.

4.    Line 136: What do you mean by “active sediments”?

5.    In terms of normalization, I was wondering why four references (UCC, NASC, BCC, and chondrite) were used. No big differences could be observed in normalized results obtained using UCC, NASC, and BCC.

6.    Line 308: Compering REEs in different river basins should be based on similarity in the hydrology and hydrogeology. Is it the case here? I would like suggest authors to do a comparison using data of rivers in the same region/country as well.  

Author Response

Response to the Reviewer #1

We greatly appreciate the first reviewer's suggestions and insightful comments. Our answers to the comments are highlighted in bright green and gray in the revised manuscript.

Comments and Suggestions for Authors

This study dealt with REE geochemistry in fluvial sediments of Lesser Zab River Basin. Based on the concentration and fractionation of REE and some other trace elements in sediments, implication for tectonic setting and provenance was discussed. I would like basically agree with the arguments presented by the authors. However, some interpretations still need to be revised, or even be changed.

Comment 1: According to the title, authors may want to explore the implication of REE geochemistry for tectonic setting and provenance. However, one cannot see any information on this objective in abstract, which should be revised accordingly.

Reply 1: We have modified the abstract and elaborated the aim of the study (lines #32-35).

Comment 2: Lines 65-66; Line 73-74. Related references should be added to support the review.

Reply 2: We modified the introduction section and have added the pertinent references accordingly (lines #80 and 91).

Comment 3: The introduction is relatively simple. It should be further enriched with more previous studies related to the research topic.

Reply 3: We have modified the introduction and added additional discussions on previous studies related to the research topic (lines #65-76 and 98-104).

Comment 4: Line 136: What do you mean by “active sediments”?

Reply 4: We used “sediments in the main course of the perennial and intermittent rivers ” instead of “active sediments” (line #156-157). 

Comment 5: In terms of normalization, I was wondering why four references (UCC, NASC, BCC, and chondrite) were used. No big differences could be observed in normalized results obtained using UCC, NASC, and BCC.

Reply 5: We think using all the four references is necessary, as elucidated in the discussion part (lines #312-318).

Comment 6: Line 308: Compering REEs in different river basins should be based on similarity in the hydrology and hydrogeology. Is it the case here? I would like suggest authors to do a comparison using data of rivers in the same region/country as well. 

Reply 6: Unfortunately, no studies were found on the geochemistry of river sediments in neighboring countries. As a result, we compared our findings with studies conducted elsewhere on the continent. Therefore, we had to clarify it in the manuscript (lines #353-359).

Reviewer 2 Report

Comments and Suggestions for Authors

Dear Editor,

The manuscript titled: “Composition of rare earth elements in fluvial sediments of Lesser Zab River Basin, Northeastern Iraq: Implication for tectonic setting and provenance"   tries to depict the provenance of fluvial sediments and unravel tectonic settings. Although the material studied, reflects a mixture, which results in many difficulties in interpreting geochemical results, I feel that the authors have successfully conducted a good discussion with some new insightful interpretations. The ms deserves to be published in the journal. It is well-written and illustrated. It can be accepted after addressing some flaws and comments. Below are appended my comments while an annotated file is also associated with my review.     

General comments  

I. In my opinion, the authors should further rework the introduction; firstly, they should introduce the inputs of REE-geochemical studies in both marine and continental settings, as well as, describe the effect of grain size on REEs. Works on REEs – bearing similar settings should be highlighted and presented in a review literature way.  The objectives, hypothesis of the study should reworked to better show the novelty of the study.

Below are some references that can help in reworking the introduction, firstly the SI in Geosciences (https://www.mdpi.com/journal/geosciences/special_issues/REEs).

- https://www.sciencedirect.com/science/article/pii/S0375674222001169

- https://www.sciencedirect.com/science/article/pii/S0169555X13005308

II. While the discussion is satisfying, the methods should be further improved by moving some parts from results and discussion to the methods such as for the methods used, including formulas.

Specific comments

1. Line 35: Why authors have only focused on the 2 mm?

2. The abstract should be reworked to include, methods and the interpretation and implication of the results. 

3. Line 83-84: Please, add further specific objectives in this chapter.

4. Line 146-151: Please identify the QC/QA and accuracy parameters.

5. Please, add standard deviation when providing mean values.

6. Some parts from the results should be profitably moved to the method section (E.g. Lines 186-188).

Author Response

Response to the Reviewer #2

We greatly appreciate the first reviewer's suggestions and insightful comments. Our answers to the comments are highlighted in turquoise and gray in the revised manuscript.

Comments and Suggestions for Authors

Dear Editor,

The manuscript titled: “Composition of rare earth elements in fluvial sediments of Lesser Zab River Basin, Northeastern Iraq: Implication for tectonic setting and provenance"   tries to depict the provenance of fluvial sediments and unravel tectonic settings. Although the material studied, reflects a mixture, which results in many difficulties in interpreting geochemical results, I feel that the authors have successfully conducted a good discussion with some new insightful interpretations. The ms deserves to be published in the journal. It is well-written and illustrated. It can be accepted after addressing some flaws and comments. Below are appended my comments while an annotated file is also associated with my review.    

General comments 

Comment 1: In my opinion, the authors should further rework the introduction; firstly, they should introduce the inputs of REE-geochemical studies in both marine and continental settings, as well as, describe the effect of grain size on REEs. Works on REEs – bearing similar settings should be highlighted and presented in a review literature way.  The objectives, hypothesis of the study should reworked to better show the novelty of the study.

Below are some references that can help in reworking the introduction, firstly the SI in Geosciences (https://www.mdpi.com/journal/geosciences/special_issues/REEs).

- https://www.sciencedirect.com/science/article/pii/S0375674222001169

- https://www.sciencedirect.com/science/article/pii/S0169555X13005308

Reply 1: We have modified the introduction and added more previous studies related to the research topic and have re-stated the objectives of the study (lines #65-76, 82-86, and 98-104).

Comment 2: While the discussion is satisfying, the methods should be further improved by moving some parts from results and discussion to the methods such as for the methods used, including formulas.

Reply 2: We have modified the manuscript by moving parts of the results chapter to the methods chapter (lines #179-191).

Specific comments

Comment 3: Line 35: Why authors have only focused on the 2 mm?

Reply 3: We have clarified the reason of focusing on the 2 mm sediments (lines #36-37, 85-86, and #160).

Comment 4: The abstract should be reworked to include, methods, interpretation, and implication of the results.

Reply 4: We modified the abstract and included the methods used, interpretation, and implication of the results (lines #32-39 and #44-47).

Comment 5: Line 83-84: Please, add further specific objectives in this chapter.

Reply 5: We added further specific objectives to the introduction (line #98-104).

Comment 6: Line 146-151: Please identify the QC/QA and accuracy parameters.

Reply 6: We identified the QC/QA and accuracy parameters in the manuscript (lines #174-179).

Comment 7: Please, add standard deviation when providing mean values.

Reply 7: We have added the standard deviation with the mean values (e.g., lines #203 and 204).

Comment 8: Some parts from the results should be profitably moved to the method section (E.g. Lines 186-188).

Reply 8: We have modified the manuscript by moving parts of the results chapter to the methods chapter (lines #179-184).

From PDF

Comment 9: Lines #36-38: Re-write

Reply 9: We rewrote the sentence (lines #37-39).

Comment 10: Line #39: which are? cite some examples

Reply 10: The sentence has been deleted.

Comment 11: Lines #53-55: Add a ref here.

Reply 11: We have added a reference (line #58).

Comment 12: Lines 75-78: Is there any effect of grain-size on the REE ? You may refer to Ferhaoui et al. 2022: Ferhaoui, S.; Kechiched, R.; Bruguier, O.; Sinisi, R.; Kocsis, L.; Mongelli, G.; Bosch, D.; Ameur-Zaimeche, O.; Laouar, R. Rare earth elements plus yttrium (REY) in phosphorites from the Tébessa region (Eastern Algeria): Abundance, geochemical distribution through grain size fractions, and economic significance. J. Geochem. Explor. 2022, 241, 107058.

Reply 11: We added sentences explaining the effect of grain-size on the REE (lines #82-86).

Comment 13: Figure 2: In the legend, you should introduce the lithology of each unit/formation (age).

Reply 13: We have added a table in the Appendix shoings the lithology and age of each unit/formation (Appendix A).

Comment 14: Line #180: why do authors have used all these standards, is there a benefit for this in interpretation.

Reply14: We think that using all the four references is necessary, as explained in the discussion part (lines #312-318).

Comment 15: Line #187: Indicate the references used to calculate these anomalies

Reply 15: Thank you for the comment. We have added the references used to calculate these anomalies (lines #179-184).

Comment 16: Table 1: add legend

Reply 16: Implemented (Table 1; lines #250-251).

Comment 17: Lines #194-197: delit

Reply 17: We deleted that.

Comment 18: Lines #239-240: I think some statistics are needed here such ANOVA -ONE WAY to check is there

Reply 18: We respect the opinion of the second reviewer. We think there is no need to do the ANOVA -ONE WAY because there is no significant difference between the studied groups.

Comment 19: Lines #273: What is the main information behind the use of all these standards?

Reply 19: We think that using all the four references is necessary. Therefore, we discussed this issue in the discussion part (lines #312-318).

Comment 20: Line #361: Eu

Reply 20: Modified (line #411).

Comment 21:Line #374: Please provide a ref for the use of this ratio

Reply 21: Implemented (line #424).

Reviewer 3 Report

Comments and Suggestions for Authors

General: The article is well written, concise and the conclusions are based on results and discussion. The references are mostly up to date and the English is fine. Some minor notes are listed below for author’s consideration.

Abstract: The main conclusions of the study need to be included in the abstract and not only descriptive results.

Introduction: Aims of the study are not mentioned in the Introduction and some statements need references (see my annotated text).

Sampling and analytical methods: You need to list all elements analysed in the study (line 146) and to report the accuracy and precision test results to validate the analysis of the samples (line 149).

Results: The term “average” is used several times. I suggest using the term “mean” instead since it is merely the sum of values divided by the number of samples. Statistically “average” implies different meaning.

Discussion: It is not wrong to compare the Lesser Zab river sediments with some Asian river sediments, but in my opinion this comparison is pointless since those rivers are crossing different and variable source rocks having different tectonic settings (line 310).

The geochemical indices used (lines 415-420) show different possibilities for source rocks: basic, basaltic, felsic, etc. In my opinion these contradicting results are an expression of the different source rocks contributing to the river sediments along its path and these rock units are of different composition as well as tectonic setting. I advise the authors to consider this when explaining the contradiction in the provenance results.

Author Response

Response to the Reviewer #3

We greatly appreciate the third reviewer's suggestions and insightful comments. Our answers to the comments are highlighted in yellow and gray in the revised manuscript.

General:

The article is well written, concise and the conclusions are based on results and discussion. The references are mostly up to date and the English is fine. Some minor notes are listed below for author’s consideration.

Comment 1: Abstract: The main conclusions of the study need to be included in the abstract and not only descriptive results.

Reply 1: We havemodified the abstract and added the main conclusions of the study (lines #37-39 and #44-47).

Comment 2: Introduction: Aims of the study are not mentioned in the Introduction and some statements need references (see my annotated text).

Reply 2: We have modified the introduction, cited the paragraphs that need references, added more previous studies related to the research topic and expanded the objectives of the study (lines #65-76 and 98-104).

Comment 3: Sampling and analytical methods: You need to list all elements analysed in the study (line 146) and to report the accuracy and precision test results to validate the analysis of the samples (line 149).

Reply 3: Our study solely focuses on examining the characteristics of REE. For this purpose, we avoided dealing with the other elements and chose selected elements for this study to support our objectives. It's worth noting that this study is one of the few that explores the geochemistry of rare earth elements in this region even though there are many studies on the trace elements. We stated that in a paragraph within the manuscript (lines #167-169).

Comment 4: Results: The term “average” is used several times. I suggest using the term “mean” instead since it is merely the sum of values divided by the number of samples. Statistically “average” implies different meaning.

Reply 4: We used the term “mean value” instead of “average” (e.g., lines #195, 196, and 209).

Comment 5: Discussion: It is not wrong to compare the Lesser Zab river sediments with some Asian river sediments, but in my opinion this comparison is pointless since those rivers are crossing different and variable source rocks having different tectonic settings (line 310).

Reply 5: Unfortunately, no studies were found on the geochemistry of river sediments in neighboring countries. Although the lithology and tectonic settings of the LZR and large rivers in Asian countries are different, their hydrology and hydrogeology are generally similar, which control REE compositions, fractionations, and distribution pattern in alluvial sediments. Due to lack of published information on REEs distribution and mobility resulting from weathering and alluvia transport for the study area, we elected to compare our results with rivers in Asian countries to gain a clear understanding of REE cycling in river sediments. Therefore, we had to clarify it in the manuscript (lines #353-359).

Comment 6: The geochemical indices used (lines 415-420) show different possibilities for source rocks: basic, basaltic, felsic, etc. In my opinion these contradicting results are an expression of the different source rocks contributing to the river sediments along its path and these rock units are of different composition as well as tectonic setting. I advise the authors to consider this when explaining the contradiction in the provenance results.

Reply 6: We have taken this comment in our consideration (lines #469-474).

From PDF

Comment 7: The abstract should contain a summary of main conclusions in addition to main results

Reply 7: We modified the abstract and expanded main conclusions of the study (lines #37-39 and #44-47).

Comment 8: You need to add the main aims and targets of this study to the Introduction

Reply 8: We added the main aims and targets of this study to the Introduction (lines 98-104).

Comment 9: Lines #53-55: Give references to these statements please

Reply 9: Implemented (line #59).

Comment 10: Lines #56-59: reference please

Reply 10: Implemented (line #62).

Comment 11: Line #146: In addition to REE please list the other elements analysed in this study

Reply 11: Our study solely focuses on examining the characteristics of REEs. For this purpose, we avoided dealing with other elements and selected specific elements to meet the needs of our study in support of our objectives. It's worth noting that this study is one of the few that explores the geochemistry of rare earth elements in this region even though there are many studies on the trace elements; this has been stated in a paragraph within the manuscript (lines #167-169).

Comment 12: Line #149: what were the results of these tests? Precision of the analysis and not the samples. You need to report the results of the precision and accuracy tests

Reply 12: We have added QC/QA and accuracy parameters in the manuscript (lines #174-179).

Comment 13: Lines #155, 156, 167, 283: mean value

Reply 13: We added the standard deviation with the mean values (e.g., lines #203 and 204).

Comment 14: Lines #164-165: references please

Reply 14: We added a reference (line #207).

Comment 15: Line #177: of the REE's in the sediments

Reply 15: Implemented (lines #219-220).

Comment 16: Line #310: Considering that these Asian rivers have various tectonic settings and different source rocks,  such comparison seems pointless.

Reply 16:  Unfortunately, no studies were found on the geochemistry of river sediments in neighboring countries. Although the lithology and tectonic settings of the LZR and large rivers in Asian countries are different, their hydrology and hydrogeology are generally similar, which control REE compositions, fractionations, and distribution pattern in alluvial sediments. Due to lack of published information on REEs distribution and mobility resulting from weathering and alluvia transport for the study area, we elected to compare our results with rivers in Asian countries to gain a clear understanding of REE cycling in river sediments. Therefore, we had to clarify it in the manuscript (lines #353-359).

Comment 17: Lines #403-404, 411, 416, 417, and 419: These contradicting results (felsic, basic, basaltic, andesitic, etc) suggest multiple source rocks which reflect the various rock units that the river crosses in its path and not necessarily due to ancient tectonic environments.

Reply 17: Thank you for your valuable comments, we have taken this comment in our consideration (lines #469-474).

Comment 17: Line #542: Correct this please

Reply 17: We corrected the reference (625).

Round 2

Reviewer 1 Report

Comments and Suggestions for Authors

No comments any more

Reviewer 2 Report

Comments and Suggestions for Authors

Dear Editor, 

Thank you for sending this new revised version of the ms. The authors have well addressed all my issues and I have no further comments. The article will constitute a contribution to REE studies of fluvial systems. The ms can be accepted for publication in the present form. I congratulate the authors.